# *LEAFY* maintains apical stem cell activity during shoot development in the fern *Ceratopteris richardii*

Andrew RG Plackett[1†‡], Stephanie J Conway[2†§], Kristen D Hewett Hazelton[2], Ester H Rabbinowitsch[1], Jane A Langdale[1*], Verónica S Di Stilio[2*]

[1]Department of Plant Sciences, University of Oxford, Oxford, United Kingdom; [2]Department of Biology, University of Washington, Seattle, United States

**Abstract** During land plant evolution, determinate spore-bearing axes (retained in extant bryophytes such as mosses) were progressively transformed into indeterminate branching shoots with specialized reproductive axes that form flowers. The LEAFY transcription factor, which is required for the first zygotic cell division in mosses and primarily for floral meristem identity in flowering plants, may have facilitated developmental innovations during these transitions. Mapping the LEAFY evolutionary trajectory has been challenging, however, because there is no functional overlap between mosses and flowering plants, and no functional data from intervening lineages. Here, we report a transgenic analysis in the fern *Ceratopteris richardii* that reveals a role for LEAFY in maintaining cell divisions in the apical stem cells of both haploid and diploid phases of the lifecycle. These results support an evolutionary trajectory in which an ancestral LEAFY module that promotes cell proliferation was progressively co-opted, adapted and specialized as novel shoot developmental contexts emerged.

DOI: https://doi.org/10.7554/eLife.39625.001

*For correspondence:
jane.langdale@plants.ox.ac.uk (JAL);
distilio@uw.edu (VSD)

†These authors contributed equally to this work

Present address: ‡Department of Plant Sciences, University of Cambridge, Cambridge, United Kigndom; §Department of Organismic and Evolutionary Biology, Harvard University, Cambridge, United States

Competing interests: The authors declare that no competing interests exist.

## Introduction

Land plants are characterized by the alternation of haploid (gametophyte) and diploid (sporophyte) phases within their lifecycle, both of which are multicellular (*Niklas and Kutschera, 2010*; *Bowman et al., 2016*). In the earliest diverging bryophyte lineages (liverworts, mosses and hornworts) the free-living indeterminate gametophyte predominates the lifecycle, producing gametes that fuse to form the sporophyte. The sporophyte embryo develops on the surface of the gametophyte, ultimately forming a simple determinate spore-producing axis (*Kato and Akiyama, 2005*; *Ligrone et al., 2012*). By contrast, angiosperm (flowering plant) sporophytes range from small herbaceous to large arborescent forms, all developing from an indeterminate vegetative shoot apex that ultimately transitions to flowering, and gametophytes are few-celled determinate structures produced within flowers (*Schmidt et al., 2015*). A series of developmental innovations during the course of land plant evolution thus simplified gametophyte form whilst increasing sporophyte complexity, with a prolonged and plastic phase of vegetative development arising in the sporophyte of all vascular plants (lycophytes, ferns, gymnosperms and angiosperms).

Studies aimed at understanding how gene function evolved to facilitate developmental innovations during land plant evolution have thus far largely relied on comparative analyses between bryophytes and angiosperms, lineages that diverged over 450 million years ago. Such comparisons have revealed examples of both sub- and neo-functionalization following gene duplication, and of co-option of existing gene regulatory networks into new developmental contexts. For example, a single bHLH transcription factor in the moss *Physcomitrella patens* regulates stomatal differentiation, whereas gene duplications have resulted in three homologs with sub-divided stomatal patterning

**eLife digest** The first plants colonized land around 500 million years ago. These plants had simple shoots with no branches, similar to the mosses that live today. Later on, some plants evolved more complex structures including branched shoots and flowers (collectively known as the "flowering plants"). Ferns are a group of plants that evolved midway between the mosses and flowering plants and have branched shoots but no flowers.

The gradual transition from simple to more complex plant structures required changes to the way in which cells divide and grow within plant shoots. Whereas animals produce new cells throughout their body, most plant cells divide in areas known as meristems. All plants grow from embryos, which contain meristems that will form the roots and shoots of the mature plant. A gene called *LEAFY* is required for cells in moss embryos to divide. However, in flowering plants *LEAFY* does not carry out this role, instead it is only required to make the meristems that produce flowers.

How did *LEAFY* transition from a general role in embryos to a more specialized role in making flowers? To address this question, Plackett, Conway et al. studied the two *LEAFY* genes in a fern called *Ceratopteris richardii*. The experiments showed that at least one of these *LEAFY* genes was active in the meristems of fern shoots throughout the lifespan of the plant. The shoots of ferns with less active *LEAFY* genes could not form the leaves seen in normal *C. richardii* plants. This suggests that as land plants evolved, the role of *LEAFY* changed from forming embryos to forming complex shoot structures.

Most of our major crops are flowering plants. By understanding how the role of *LEAFY* has changed over the evolution of land plants, it might be possible to manipulate *LEAFY* genes in crop plants to alter shoot structures to better suit specific environments.

DOI: https://doi.org/10.7554/eLife.39625.002

roles in the angiosperm *Arabidopsis thaliana* (hereafter 'Arabidopsis') (**MacAlister and Bergmann, 2011**); class III HD-ZIP transcription factors play a conserved role in the regulation of leaf polarity in *P. patens* and Arabidopsis but gene family members have acquired regulatory activity in meristems of angiosperms (**Yip et al., 2016**); and the gene regulatory network that produces rhizoids on the gametophytes of both the moss *P. patens* and the liverwort *Marchantia polymorpha* has been co-opted to regulate root hair formation in Arabidopsis sporophytes (**Menand et al., 2007**; **Pires et al., 2013**; **Proust et al., 2016**). In many cases, however, interpreting the evolutionary trajectory of gene function by comparing lineages as disparate as bryophytes and angiosperms has proved challenging, particularly when only a single representative gene remains in most extant taxa – as is the case for the *LEAFY* (*LFY*) gene family (**Himi et al., 2001**; **Maizel et al., 2005**; **Sayou et al., 2014**).

The LFY transcription factor, which is present across all extant land plant lineages and related streptophyte algae (**Sayou et al., 2014**), has distinct functional roles in bryophytes and angiosperms. In *P. patens*, LFY regulates cell divisions during sporophyte development (including the first division of the zygote) (**Tanahashi et al., 2005**), whereas in angiosperms the major role is to promote the transition from inflorescence to floral meristem identity (**Carpenter and Coen, 1990**; **Schultz, 1991**; **Weigel et al., 1992**; **Blázquez et al., 1997**; **Souer et al., 1998**; **Molinero-Rosales et al., 1999**). Given that LFY proteins from liverworts and all vascular plant lineages tested to date (ferns, gymnosperms and angiosperms) bind a conserved target DNA motif, whereas hornwort and moss homologs bind to different lineage-specific motifs (**Sayou et al., 2014**), the divergent roles in mosses and angiosperms may have arisen through the activation of distinct networks of downstream targets. This suggestion is supported by the observation that PpLFY cannot complement loss-of-function *lfy* mutants in Arabidopsis (**Maizel et al., 2005**). Similar complementation studies indicate progressive functional changes as vascular plant lineages diverged in that the *lfy* mutant is not complemented by lycophyte LFY proteins (**Yang et al., 2017**) but is partially and progressively complemented by fern and gymnosperm homologs (**Maizel et al., 2005**). Because LFY proteins from ferns, gymnosperms and angiosperms recognize the same DNA motif, this progression likely reflects co-option of an ancestral *LFY* gene regulatory network into different developmental contexts. As such, the role in floral meristem identity in angiosperms would have been co-opted from an unknown ancestral

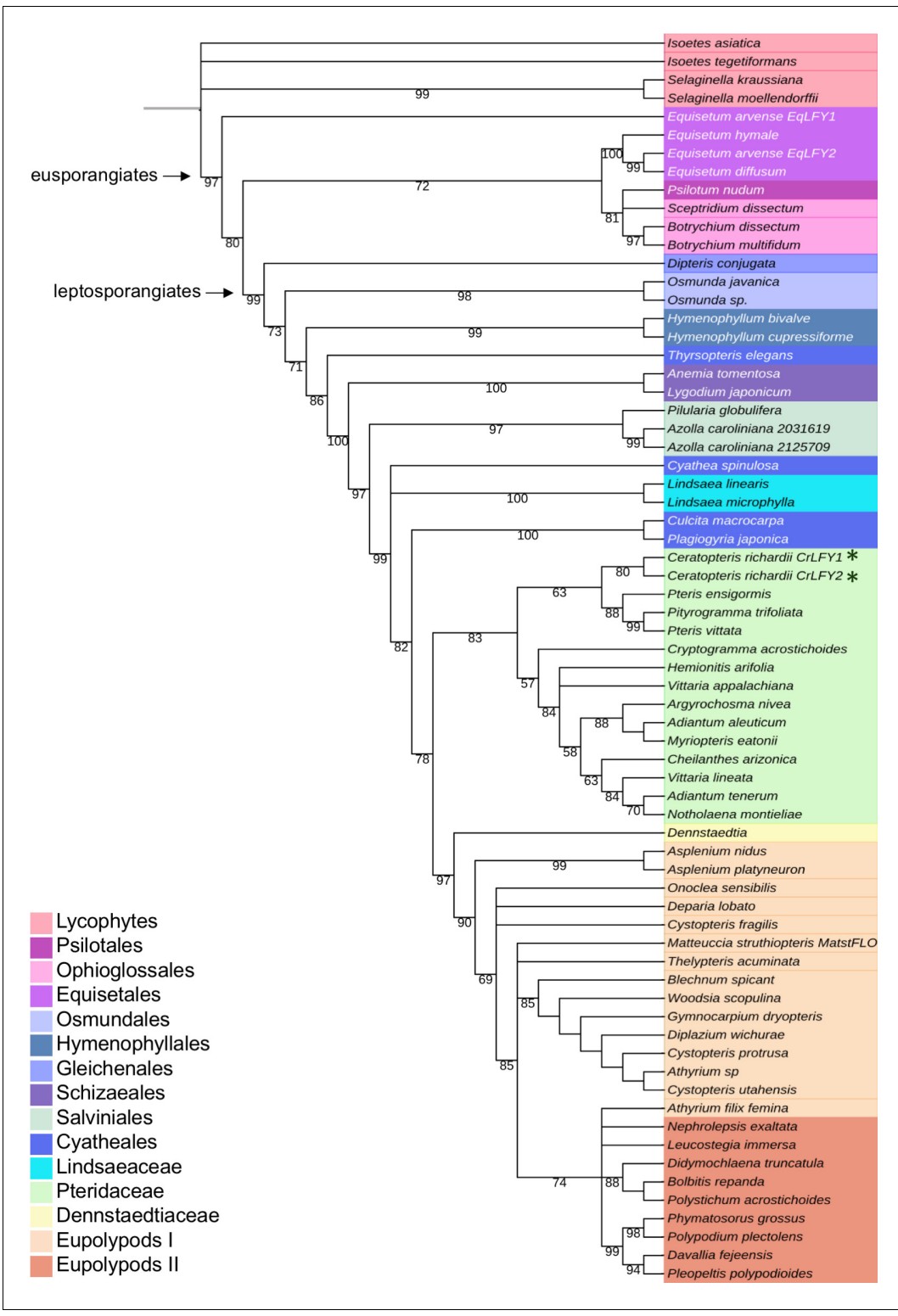

**Figure 1.** *CrLFY1* and *CrLFY2* arose from a recent gene duplication event. Inferred phylogenetic tree from maximum likelihood analysis of 64 LFY amino acid sequences (see *Supplementary file 1* for accession numbers) sampled from within the fern lineage plus lycophyte sequences as an outgroup. Bootstrap values are given for each node. The tree shown is extracted from a phylogeny with representative sequences from all land plant lineages (*Figure 1—figure supplement 1*). The *Ceratopteris richardii* genome contains no more than two copies of LFY (*Figure 1—figure supplement 2*; indicated by *). Different taxonomic clades within the fern lineage are

*Figure 1 continued on next page*

*Figure 1 continued*
denoted by different colours, as shown. The divergence between eusporangiate and leptosporangiate ferns is indicated by arrows.
DOI: https://doi.org/10.7554/eLife.39625.003
The following figure supplements are available for figure 1:
**Figure supplement 1.** Phylogenetic relationships between *LEAFY* sequences reflect established relationships within vascular plant lineages.
DOI: https://doi.org/10.7554/eLife.39625.004
**Figure supplement 2.** The *Ceratopteris* genome contains only two copies of *LFY*.
DOI: https://doi.org/10.7554/eLife.39625.005

context in non-flowering vascular plants, a context that cannot be predicted from existing bryophyte data.

The role of *LFY* in non-flowering vascular plant lineages has thus far been hypothesized on the basis of expression patterns in the lycophyte *Isoetes sinensis* (*Yang et al., 2017*), several gymnosperm species (*Mellerowicz et al., 1998*; *Mouradov et al., 1998*; *Shindo et al., 2001*; *Carlsbecker et al., 2004*; *Vázquez-Lobo et al., 2007*; *Carlsbecker et al., 2013*) and the fern *Ceratopteris richardii* (hereafter 'Ceratopteris') (*Himi et al., 2001*), which has been used as a model of fern development for a number of years (*Hickok et al., 1995*). These studies reported broad expression in vegetative and reproductive sporophyte tissues of *I. sinensis* and gymnosperms, and in both gametophytes and sporophytes of Ceratopteris. Although gene expression can be indicative of potential roles in each case, the possible evolutionary trajectories and differing ancestral functions proposed for *LFY* within the vascular plants (*Theissen and Melzer, 2007*; *Moyroud et al., 2010*) cannot be resolved without functional validation. Here we present a functional analysis in Ceratopteris that reveals a stem cell maintenance role for at least one of the two *LFY* homologs in both gametophyte and sporophyte shoots and discuss how that role informs our mechanistic understanding of developmental innovations during land plant evolution.

## Results

### The *CrLFY1* and *CrLFY2* genes duplicated recently within the fern lineage

The *LFY* gene family is present as a single gene copy in most land plant genomes (*Sayou et al., 2014*). In this regard, the presence of two *LFY* genes in Ceratopteris (*Himi et al., 2001*) is atypical. To determine whether this gene duplication is more broadly represented within the ferns and related species (hereafter 'ferns'), a previous amino acid alignment of LFY orthologs (*Sayou et al., 2014*) was pruned and supplemented with newly-available fern homologs (see Materials and methods) to create a dataset of 120 sequences,~50% of which were from the fern lineage (*Supplementary file 1–3*). The phylogenetic topology inferred within the vascular plants using the entire dataset (*Figure 1—figure supplement 1*) was consistent with previous analyses (*Qiu et al., 2006*; *Wickett et al., 2014*). Within the ferns (64 in total), phylogenetic relationships between *LFY* sequences indicated that the two gene copies identified in *Equisetum arvense*, *Azolla caroliniana* and Ceratopteris each resulted from recent independent duplication events (*Figure 1*). Gel blot analysis confirmed the presence of no more than two *LFY* genes in the Ceratopteris genome (*Figure 1—figure supplement 2*). Given that the topology of the tree excludes the possibility of a gene duplication prior to diversification of the ferns, *CrLFY1* and *CrLFY2* are equally orthologous to the single copy *LFY* representatives in other fern species.

### *CrLFY1* and *CrLFY2* transcripts accumulate differentially during the Ceratopteris lifecycle

The presence of two *LFY* genes in the Ceratopteris genome raises the possibility that gene activity was neo- or sub-functionalized following duplication. To test this hypothesis, transcript accumulation patterns of *CrLFY1* and *CrLFY2* were investigated throughout the Ceratopteris lifecycle (shown as a schematic in *Figure 2* for reference).

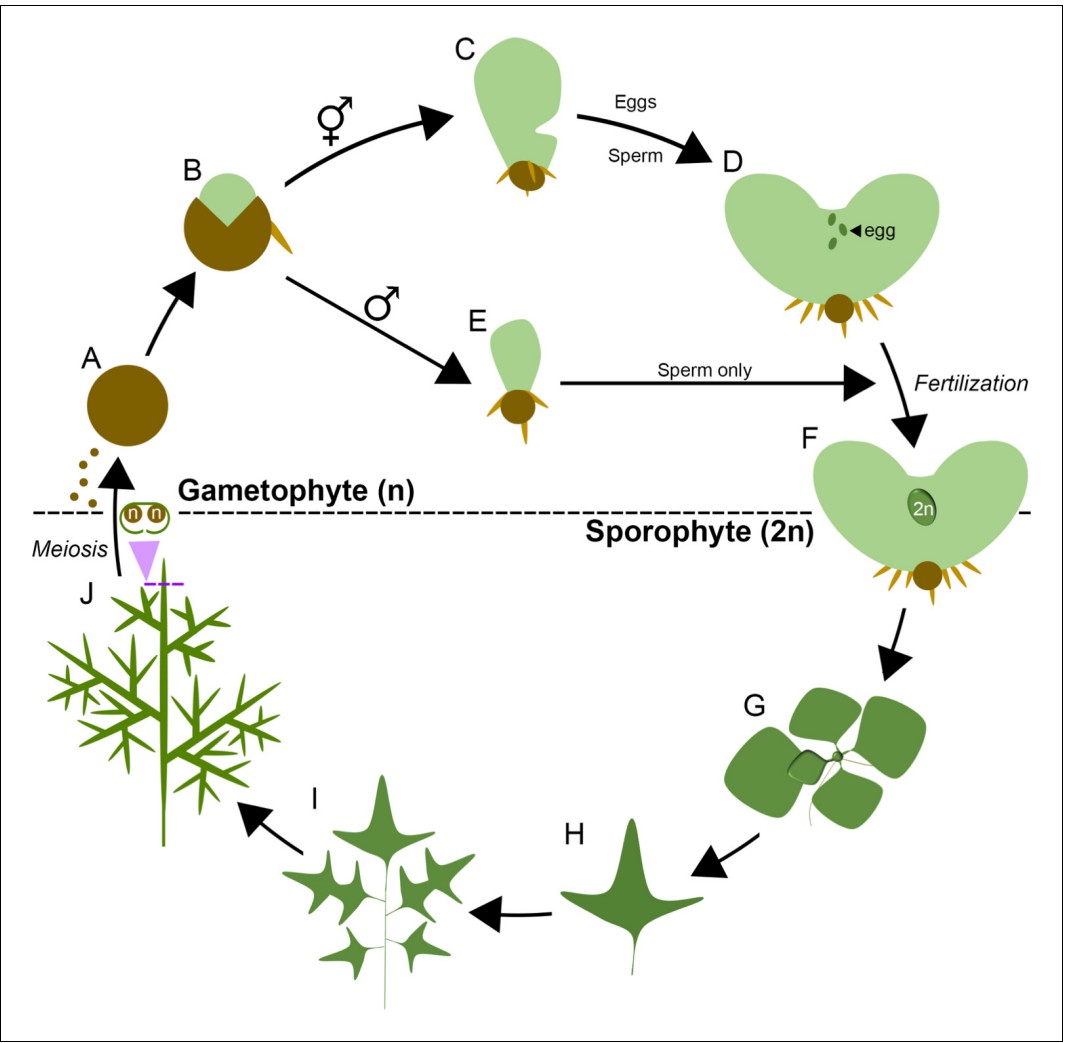

**Figure 2.** The lifecycle of *Ceratopteris richardii*. Ceratopteris propagates in the haploid gametophyte phase of its lifecycle (n) through single-celled spores (**A**) On spore germination (**B**) a two-dimensional photosynthetic thallus develops into one of two sexes, a default hermaphrodite (**C**) which produces eggs and sperm (**D**) or a hormone-induced male that produces sperm only (**E**). Eggs are retained on the hermaphrodite thallus, and fertilization results in the development of a diploid (2n) embryo on the gametophyte (**F**), initiating the sporophyte phase of the lifecycle. The sporophyte establishes a vegetative shoot that initiates leaflike lateral organs (fronds) and roots from its apex (**G**). The first fronds produced are simple but later fronds become increasingly lobed and dissected (**H, I**). The sporophyte undergoes a reproductive phase-change and subsequent fronds generate haploid spores by meiosis on their undersides (**J**), enclosed in a morphologically-distinct curled lamina. Mature spores are dispersed to restart the lifecycle.

DOI: https://doi.org/10.7554/eLife.39625.006

The developmental stages sampled spanned from imbibed spores prior to germination of the haploid gametophyte (*Figure 3A*), to differentiated male and hermaphrodite gametophytes (*Figure 3B–D*), through fertilization and formation of the diploid sporophyte embryo (*Figure 3E*), to development of the increasingly complex sporophyte body plan (*Figure 3F–K*). Quantitative real-time PCR (qRT-PCR) analysis detected transcripts of both *CrLFY1* and *CrLFY2* at all stages after spore germination, but only *CrLFY2* transcripts were detected in spores prior to germination (*Figure 3L*). A two-way ANOVA yielded a highly significant interaction (F(10,22) = 14.21; p<0.0001) between gene copy and developmental stage that had not been reported in earlier studies (*Himi et al., 2001*), and is indicative of differential gene expression between *CrLFY1* and *CrLFY2* that is dependent on developmental stage. Of particular note were significant differences between

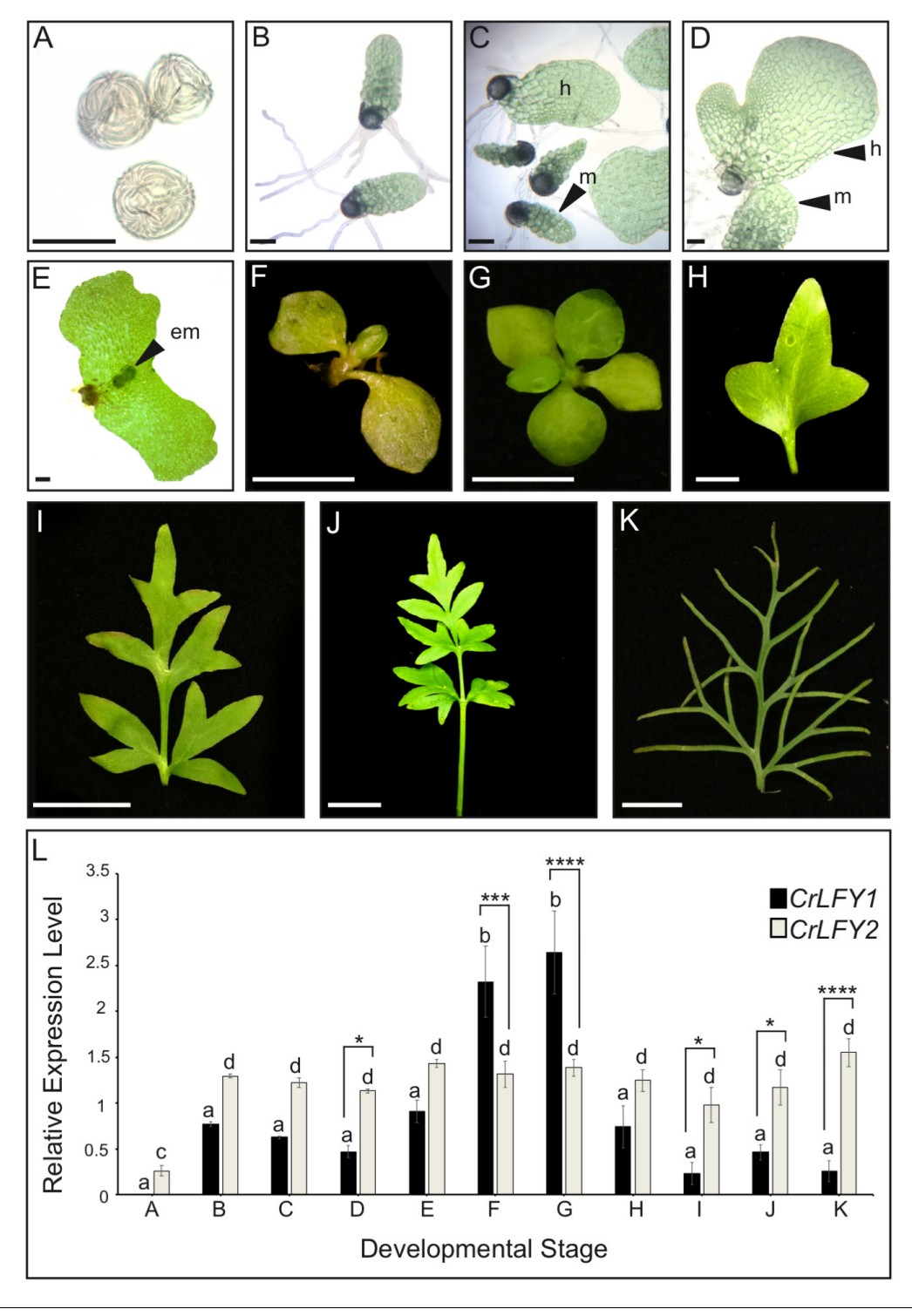

Figure 3. *CrLFY1* and *CrLFY2* are differentially expressed during the Ceratopteris lifecycle. (A-K) Representative images of the developmental stages sampled for expression analysis in (L). Imbibed spores (A); populations of developing gametophytes harvested at 5 (B, C) and 8 (D) days after spore-sowing (DPS), comprising only males (B) or a mixture of hermaphrodites (h) and males (m) (C, D); fertilized gametophyte subtending a developing sporophyte embryo (em) (E); whole sporophyte shoots comprising the shoot apex with 3 (F) or five expanded entire fronds attached (G); individual vegetative fronds demonstrating a heteroblastic progression in which frond complexity increases through successive iterations of lateral outgrowths (pinnae) (H–J); complex fertile frond with
*Figure 3 continued on next page*

*Figure 3 continued*

sporangia on the underside of individual pinnae (**K**). Scale bars = 100 um (**A–E**), 5 mm (**F–H**), 20 mm (**I–K**). (**L**) Relative expression levels of *CrLFY1* and *CrLFY2* (normalized against the housekeeping genes *CrACTIN1* and *CrTBP*) at different stages of development. *n* = 3; Error bars = standard error of the mean (SEM). Pairwise statistical comparisons (ANOVA followed by Tukey's multiple comparisons test– **Supplementary file 4**) found no significant difference in *CrLFY2* transcript levels between any gametophyte or sporophyte tissues sampled after spore germination ($p > 0.05$) and no significant difference between *CrLFY1* and *CrLFY2* transcript levels during early gametophyte development ($p > 0.05$) (**B, C**). Differences between *CrLFY1* and *CrLFY2* transcript levels were significant in gametophytes at 8 DPS ($p < 0.05$) (**D**). *CrLFY1* transcript levels were significantly higher in whole young sporophytes (**F**) and vegetative shoots (**G**) compared to isolated fronds (**H–K**) ($p < 0.05$). *CrLFY1* transcript levels in whole sporophytes and shoots were greater than *CrLFY2*, whereas in isolated fronds *CrLFY1* transcript levels were consistently lower than *CrLFY2* ($p < 0.05$). Asterisks denote significant difference (*, $p < 0.05$; **, $p < 0.01$, ***, $p < 0.001$; ****, $p < 0.0001$) between *CrLFY1* and *CrLFY2* transcript levels (Sidak's multiple comparisons test) within a developmental stage. Letters denote significant difference ($p < 0.05$) between developmental stages for *CrLFY1* or *CrLFY2* (Tukey's test). Groups marked with the same letter are not significantly different from each other ($p > 0.05$). Statistical comparisons between developmental stages were considered separately for *CrLFY1* and *CrLFY2*. The use of different letters between *CrLFY1* and *CrLFY2* does not indicate a significant difference.
DOI: https://doi.org/10.7554/eLife.39625.007
The following source data is available for figure 3:

**Source data 1.** CrLFY qRT-PCR ontogenic expression data
DOI: https://doi.org/10.7554/eLife.39625.008

---

*CrLFY1* and *CrLFY2* transcript levels during sporophyte development (**Supplementary file 4**). Whereas *CrLFY2* transcript levels were similar across sporophyte samples, *CrLFY1* transcript levels were much higher in samples that contained the shoot apex (**Figure 3F,G**) than in those that contained just fronds (**Figure 3H–K**). These data suggest that *CrLFY1* and *CrLFY2* genes may play divergent roles during sporophyte development, with *CrLFY1* acting primarily in the shoot apex and *CrLFY2* acting more generally.

## Spatial expression patterns of *CrLFY1* are consistent with a retained ancestral role to facilitate cell divisions during embryogenesis

Functional characterization in *P. patens* previously demonstrated that PpLFY promotes cell divisions during early sporophyte development (**Tanahashi et al., 2005**). To determine whether the spatial domains of *CrLFY1* expression are consistent with a similar role in Ceratopteris embryo (early sporophyte) development, transgenic lines were generated that expressed the reporter gene B-glucuronidase (GUS) driven by a 3.9 kb fragment of the *CrLFY1* promoter (*CrLFY1_{pro}::GUS*). This promoter fragment comprised genomic sequence encoding the entire published 5'UTR (**Himi et al., 2001**) plus a further 1910 bp upstream of the predicted transcription start site (**Figure 1—figure supplement 2A**). In the absence of a genome sequence, repeated attempts to isolate an analogous fragment of *CrLFY2* sequence were unsuccessful (see Materials and methods for details). Construct maps plus DNA blot and PCR validation of transgenic lines are shown in **Figure 4—figure supplements 1–4**. GUS activity was monitored in individuals from three independent transgenic lines, sampling both before and up to six days after fertilization (**Figure 4A–O**), using wild-type individuals as negative controls (**Figure 4P–T**) and individuals from a transgenic line expressing GUS driven by the constitutive 35S promoter (*35S_{pro}*) as positive controls (**Figure 4U–Y**). Notably, no GUS activity was detected in unfertilized archegonia of *CrLFY1_{pro}::GUS* gametophytes (**Figure 4A,F,K**) but by two days after fertilization (DAF) GUS activity was detected in most cells of the early sporophyte embryo (**Figure 4B,G,L**). At 4 DAF, activity was similarly detected in all visible embryo cells, including the embryonic frond, but not in the surrounding gametophytic tissue (the calyptra) (**Figure 4C,H,M**). This embryo-wide pattern of GUS activity became restricted in the final stages of development such that by the end of embryogenesis (6 DAF) GUS activity was predominantly localized in the newly-initiated shoot apex (**Figure 4D,E,I,J,N,O**). Collectively, the GUS activity profiles indicate that *CrLFY1* expression is induced following formation of the zygote, sustained in cells of the embryo that are actively dividing, and then restricted to the shoot apex at embryo maturity. This profile is consistent with the suggestion that *CrLFY1* has retained the LFY role first identified in *P. patens*

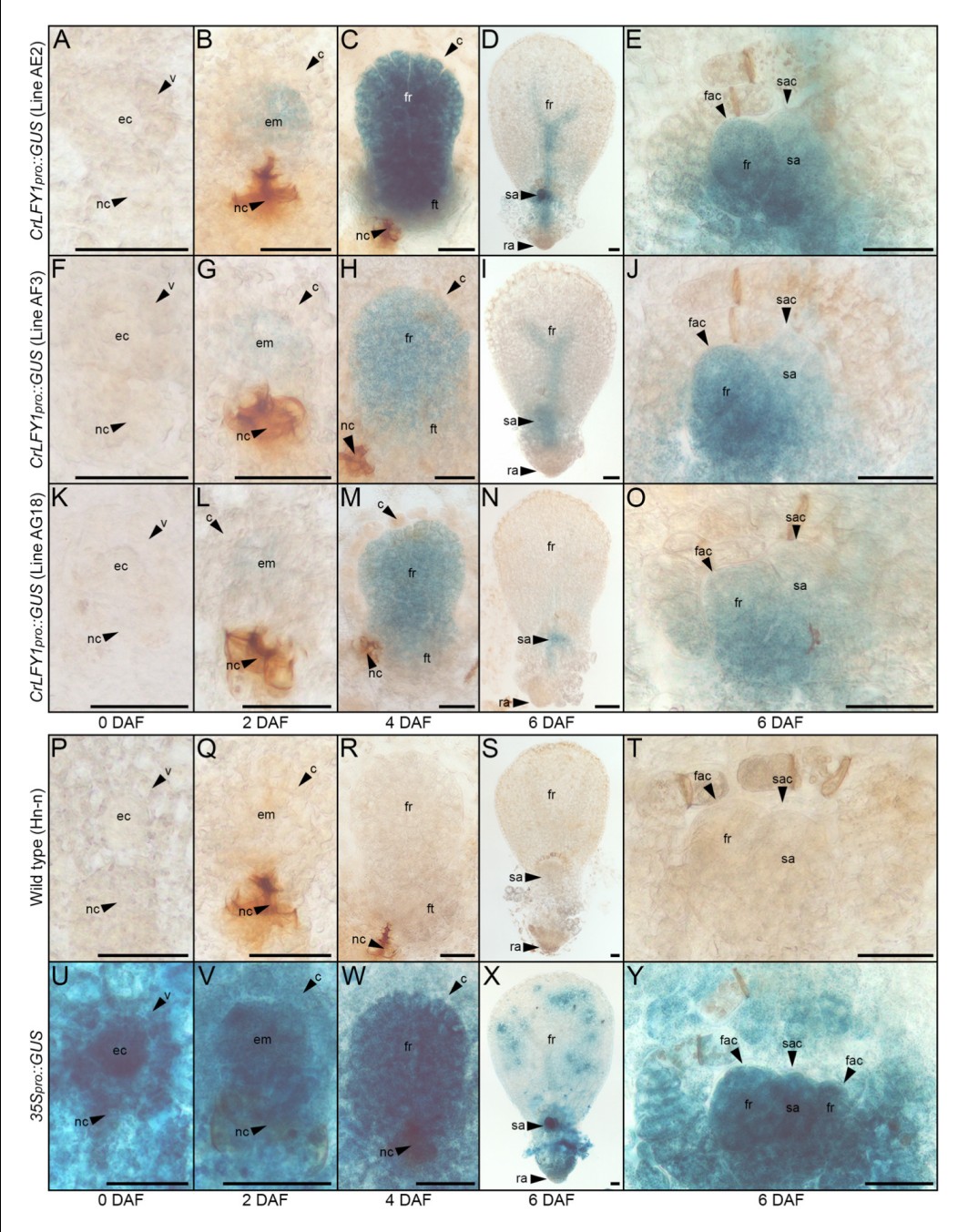

**Figure 4.** The *CrLFY1* promoter drives reporter gene expression in proliferating tissues of the developing Ceratopteris embryo. (A–Y) GUS activity detected as blue staining in developing embryos of three independent *CrLFY1pro::GUS* transgenic reporter lines (A–O), a representative negative wild-type control line (P–T) and a representative positive *35Spro::GUS* control line (U–Y). Tissues are shown prior to fertilization (A, F, K, P, U), or 2 (B, G, L, Q, V), 4 (C, H, M, R, W), and 6 (D, I, N, S, X) days after fertilization (DAF). In *CrLFY1pro::GUS* lines, GUS activity first became visible within the first few divisions of embryo development (but not in surrounding gametophyte tissues) at 2 DAF (B, G, L) and was expressed in cells of the embryo frond as it proliferated (C, H, M). GUS activity was visible in the shoot apex and in frond vascular tissue at 6 DAF (D, I, N), with staining in the shoot apical cell (sac), subtending shoot apex tissues and newly-initiated fronds, including the frond apical cell (fac) (E, J, O). No GUS activity was detected in wild-type samples (P–T), whereas the majority of cells in the constitutively expressing *35Spro::GUS* samples stained blue (U–Y). Embryos develop on the surface of the gametophyte thallus when an egg cell (ec) within the archegonium (which comprises a venter (v) and neck cells (nc) to allow sperm

*Figure 4 continued on next page*

*Figure 4 continued*
entry) are fertilized. After fertilization, the venter forms a jacket of haploid cells known as the calyptra (c) that surrounds the diploid embryo (em). Cell fates in the embryo (embryo frond (fr), embryo foot (ft), root apex (ra) and shoot apex (sa)) are established at the eight-celled stage (*Johnson and Renzaglia, 2008*), which is around 2 DAF under our growth conditions. Embryogenesis is complete at 6 DAF, after which fronds arise from the shoot apex. Scale bars = 50 μm.
DOI: https://doi.org/10.7554/eLife.39625.009
The following figure supplements are available for figure 4:
**Figure supplement 1.** Schematic of *CrLFY1pro::GUS* and *35S$_{pro}$::GUS* constructs.
DOI: https://doi.org/10.7554/eLife.39625.010
**Figure supplement 2.** DNA gel blot analysis of *CrLFY1$_{pro}$::GUS* and *35S$_{pro}$::GUS* transgenic lines.
DOI: https://doi.org/10.7554/eLife.39625.011
**Figure supplement 3.** PCR analysis of *CrLFY1$_{pro}$::GUS* T$_1$ lines identified full-length or near full-length *CrLFY1* promoter sequences in T-DNA insertions.
DOI: https://doi.org/10.7554/eLife.39625.012
**Figure supplement 4.** PCR analysis of *35S$_{pro}$::GUS* positive control line identified a full-length *35S$_{pro}$::GUS* insertion.
DOI: https://doi.org/10.7554/eLife.39625.013

(*Tanahashi et al., 2005*), namely to promote the development of a multicellular sporophyte, in part by facilitating the first cell division of the zygote.

## *CrLFY1* is expressed in dividing tissues throughout shoot development

Both mosses and ferns form embryos, but moss sporophyte development is determinate post-embryogenesis (*Kato and Akiyama, 2005*; *Kofuji and Hasebe, 2014*) whereas fern sporophytes are elaborated post-embryonically from indeterminate shoot apices (*Bierhorst, 1977*; *White and Turner, 1995*). The Ceratopteris shoot apex comprises a single apical cell that generates daughter cells through asymmetric divisions, and individual lateral organs (fronds and root) arise from their own apical cells specified within the grouped descendants of these daughter cells (*Hou and Hill, 2002*; *Hou and Hill, 2004*). *CrLFY1$_{pro}$::GUS* expression in the shoot apex at the end of embryogenesis (*Figure 4E,J,O*) and elevated transcript levels in shoot apex-containing sporophyte tissues (*Figure 3L*) suggested an additional role for *CrLFY1* relative to that seen in mosses, namely to promote proliferation in the indeterminate shoot apex. To monitor *CrLFY1* expression patterns in post-embryonic sporophytes, GUS activity was assessed in *CrLFY1$_{pro}$::GUS* lines at two stages of vegetative development (*Figure 5A–O*) and after the transition to reproductive frond formation (*Figure 5—figure supplement 1A–L*). Wild-type individuals were used as negative controls (*Figure 5P–T*; *Figure 5—figure supplement 1M–P*) and *35S$_{pro}$::GUS* individuals as positive controls (*Figure 5U–Y*; *Figure 5—figure supplement 1Q–T*). In young sporophytes (20 DAF), GUS activity was primarily localized in shoot apical tissues and newly-emerging frond primordia (*Figure 5A,F,K*), with very little activity detected in the expanded simple fronds produced at this age (*Figure 5B,G,L*). In older vegetative sporophytes (60 DAF), which develop complex dissected fronds (*Figure 5C,H,M*), GUS activity was similarly localized in the shoot apex and young frond primordia in two out of the three fully characterized lines (*Figure 5D,I,N*) and in a total of 8 out of 11 lines screened (from seven independent rounds of plant transformation). GUS activity was also detected in developing fronds in regions where the lamina was dividing to generate pinnae and pinnules (*Figure 5E,J,O*). In some individuals GUS activity could be detected in frond tissues almost until maturity (*Figure 5C*). Notably, patterns of *CrLFY1$_{pro}$::GUS* expression were the same in the apex and complex fronds of shoots before (60 DAF) (*Figure 5C–E,H–J,M–O*) and after (~115 DAF) the reproductive transition (*Figure 5—figure supplement 1A–L*). Consistent with a general role for *CrLFY1* in promoting cell proliferation in the shoot, GUS activity was also detected in shoot apices that initiate *de novo* at the lamina margin between pinnae (*Figure 5Z–AD*). Together these data support the hypothesis that *LFY* function was recruited to regulate cell division processes in the shoot when sporophytes evolved from determinate to indeterminate structures.

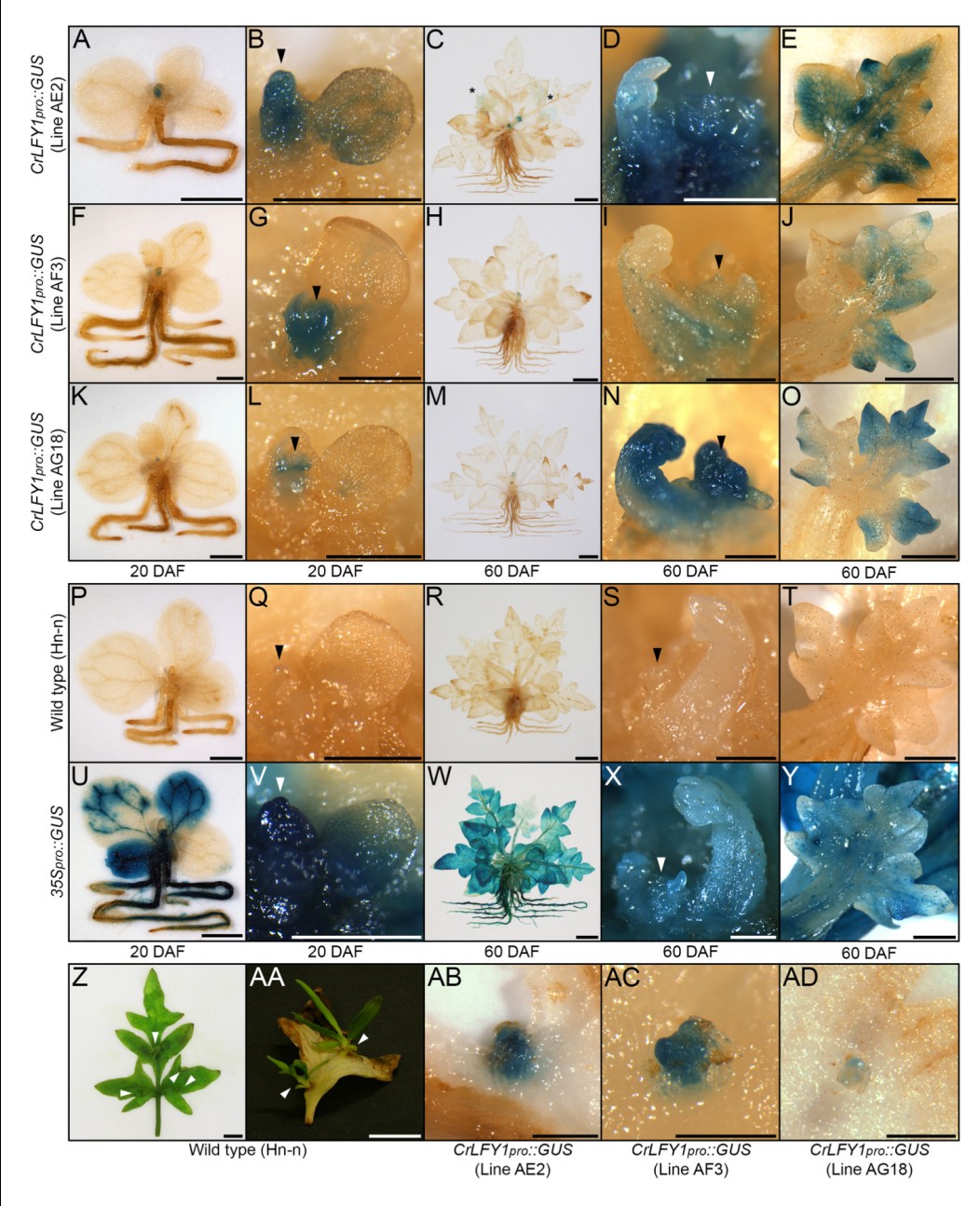

**Figure 5.** The *CrLFY1* promoter drives reporter gene expression in proliferating shoot tissues of the Ceratopteris sporophyte. (A–Y) GUS activity detected as blue staining in post-embryonic sporophytes from three independent *CrLFY1pro::GUS* transgenic reporter lines (A–O), negative wild-type controls (P–T) and positive *35Spro::GUS* controls (U–Y). Sporophytes were examined at 20 DAF (A, B, F, G, K, L, P, Q, U, V) and 60 DAF (C–E, H–J, M–O, R–T, W–Y). GUS staining patterns are shown for whole sporophytes (A, C, F, H, K, M, P, R, U, W), shoot apices (arrowheads) (B, D, G, I, L, N, Q, S, V, X) and developing fronds (E, J, O, T, Y). In *CrLFY1pro::GUS* sporophytes at 20 DAF (producing simple, spade-like fronds) GUS activity was restricted to the shoot apex (A, F, K) and newly-initiated frond primordia, with very low activity in expanded fronds (B, G, L). In *CrLFY1pro::GUS* sporophytes at 60 DAF (producing complex, highly dissected fronds) GUS activity was similarly seen in the apex (C, H, M), but persisted for longer during frond development. Activity was initially detected throughout the frond primordium (D, I, N), before becoming restricted to actively proliferating areas of the lamina (E, J, O). Scale bars = 2 mm (A, F, K, P, U), 500 µm (B, D, G, I, L, N, Q, S, V, X) 10 mm (C, H, M, R, W), 1 mm (E, J, O, T, Y). *=GUS staining in maturing frond. GUS staining patterns were the same in leaves formed after the reproductive transition (*Figure 5—figure supplement 1*). (Z-AD) Fronds can initiate *de novo* shoots (white arrowheads) from marginal tissue between

*Figure 5 continued on next page*

*Figure 5 continued*

existing frond pinnae (**Z, AA**). GUS activity was detected in emerging de novo shoot apices on *CrLFY1_pro::GUS* fronds (**AB–AD**). Scale bars = 10 mm (**Z, AA**), 500 μm (**AB–AD**).

DOI: https://doi.org/10.7554/eLife.39625.014

The following figure supplement is available for figure 5:

**Figure supplement 1.** *CrLFY1_pro::GUS* expression patterns are similar in Ceratopteris shoots before and after reproductive phase change.

DOI: https://doi.org/10.7554/eLife.39625.015

## *CrLFY1* regulates activity of the sporophyte shoot apex

To test the functional significance of *CrLFY* expression patterns, transgenic RNAi lines were generated in which one of four RNAi constructs targeted to *CrLFY1*, *CrLFY2* or both were expressed from the maize ubiquitin promoter (*ZmUbi_pro*). Construct maps plus DNA blot and PCR validation of transgenic lines are shown in **Figure 6—figure supplements 1–5**. Genotypic screening identified 10 lines

**Table 1.** Summary of *CrLFY* RNAi transgenic lines and their phenotypic characterization.

Transgenic lines exhibited gametophytic developmental arrest and/or sporophyte shoot termination at varying stages of development. '+' indicates that a particular line was phenotypically normal at the developmental stage indicated, '−' indicates that development had arrested at or prior to this stage. In lines marked '+/-' the stage at which developmental defects occurred varied between individuals within the line, and at least some arrested individuals were identified at the stage indicated. The five *ZmUbi_pro::CrLFY1/2-i1* lines shown were generated from three rounds of transformation, the pairs of lines B16 and B19 and D2 and D4 potentially arising from the same transformation event. The no hairpin control lines NHC-2 (F3) and NHC-3 (F4) may similarly have arisen from a single transformation event. In all other cases, each transgenic line arose from a separate round of transformation and so must represent independent T-DNA insertions.

| RNAi transgene | Line | Transformation replicate | Gametophyte phase | | | | Sporophyte phase | | | | |
| | | | Spore germin-ation | AC-based growth | Notch meristem-based growth | % arrested | Embryo | Shoot apex initiated | Simple frond | Complex frond | % arrested |
| --- | --- | --- | --- | --- | --- | --- | --- | --- | --- | --- | --- |
| *ZmUBI_pro::CrLFY1/2-i1* | B16 | 1 | + | - | - | 99.86 | + | + | - | - | <5% |
| *ZmUBI_pro::CrLFY1/2-i1* | B19 | 1 | + | - | - | 50.00 | + | + | + | - | <5% |
| *ZmUBI_pro::CrLFY1/2-i1* | D13 | 2 | + | - | - | 99.80 | + | + | - | - | <5% |
| *ZmUBI_pro::CrLFY1/2-i1* | D2 | 3 | + | + | + | 0.00 | + | + | - | - | <5% |
| *ZmUBI_pro::CrLFY1/2-i1* | D4 | 3 | + | + | + | 0.00 | + | + | + | + | <5% |
| *ZmUBI_pro::CrLFY1/2-i2* | F9 | 4 | + | - | - | 0.00 | + | + | - | - | <5% |
| *ZmUBI_pro::CrLFY1/2-i2* | F14 | 5 | - | - | - | 100.00 | - | - | - | - | 0 |
| *ZmUBI_pro::CrLFY1-i3* | E8 | 6 | + | +/- | +/- | 100.00 | - | - | - | - | 0 |
| *ZmUBI_pro::CrLFY1-i3* | G13 | 7 | + | + | + | 0.00 | + | + | - | - | <5% |
| *ZmUBI_pro::CrLFY2-i4* | C3 | 8 | + | + | + | 0.00 | + | + | - | - | <5% |
| NHC-1 (control) | D20 | 9 | + | + | + | 0.00 | + | + | + | + | 0 |
| NHC-2 (control) | F3 | 10 | + | + | + | 0.00 | + | + | + | + | 0 |
| NHC-3 (control) | F4 | 10 | + | + | + | 0.00 | + | + | + | + | 0 |

DOI: https://doi.org/10.7554/eLife.39625.016

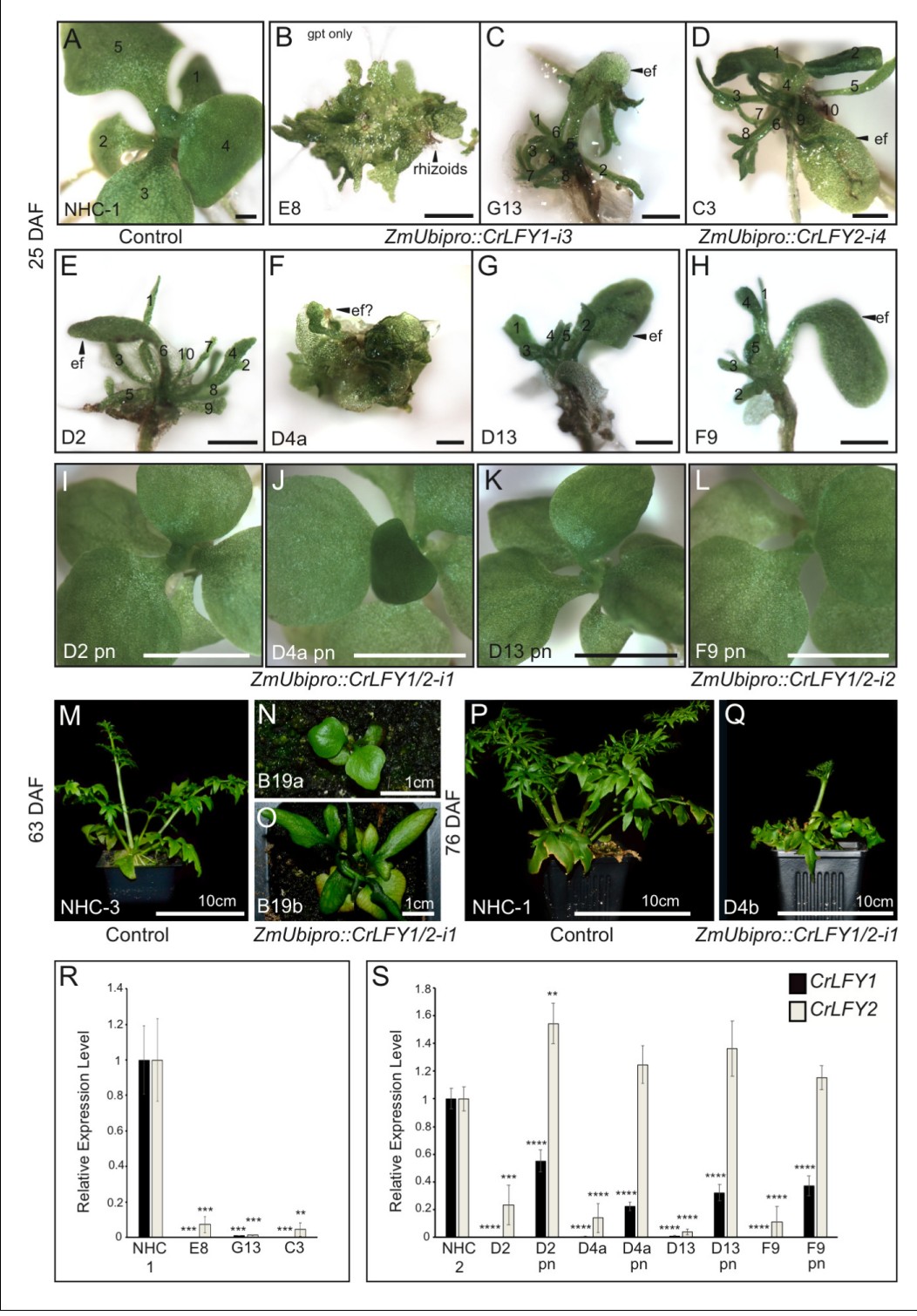

**Figure 6.** Suppression of *CrLFY* expression causes early termination of the Ceratopteris sporophyte shoot apex. (A-L) Sporophyte phenotype 25 days after fertilization (DAF) in no hairpin control, NHC-1 (A) and transgenic lines carrying RNAi constructs against *CrLFY1* (*ZmUbi_pro_::CrLFY1-i3*) (B, C), *CrLFY2* (*ZmUbi_pro_::CrLFY1-i4*) (D) and both *CrLFY1* and *CrLFY2* (*ZmUbi_pro_::CrLFY1/2-i1* and *ZmUbi_pro_::CrLFY1/2-i2*) (E–L). In some lines, both aborted and phenotypically normal sporophytes were identified (compare E and I; F and J; G and K; H and L). The presence of the RNAi transgene in phenotypically normal sporophytes was validated by genotyping (*Figure 6—figure supplement 5*). Scale bars = 1 mm (A–H), 5 mm (I–L). (M–Q) Sporophyte phenotype of two no hairpin control

*Figure 6 continued on next page*

*Figure 6 continued*

(NHC-3 and NHC-1) (**M, P**) and two *ZmUbi$_{pro}$::CrLFY1/2-i1* (**N, O**) lines at 63 (**M–O**) and 76 (**P,Q**) DAF. (**R, S**) qRT-PCR analysis of *CrLFY1* and *CrLFY2* transcript levels (normalized against the averaged expression of reference genes *CrACTIN1* and *CrTBP*) in the sporophytes of the RNAi lines shown in (**A–L**). Transcript levels are depicted relative to no hairpin controls (NHC-1or −3), *n* = 3, error bars = standard error of the mean (SEM). *CrLFY1* and *CrLFY2* expression levels were significantly reduced compared to controls ($p<0.01$ or less) in all transgenic lines where sporophyte shoots undergo early termination (**A–H**), but in phenotypically normal (pn) sporophytes segregating in the same lines (**I–L**), only *CrLFY1* transcript levels were reduced ($p<0.0001$). *CrLFY2* transcript levels in pn sporophytes were not significantly lower than in controls. Asterisks denote level of significant difference from controls (**$p<0.01$, ***$p<0.001$; ****$p<0.0001$).
DOI: https://doi.org/10.7554/eLife.39625.017

The following source data and figure supplements are available for figure 6:

**Source data 1.** CrLFY RNAi lines qRT-PCR expression data
DOI: https://doi.org/10.7554/eLife.39625.023
**Figure supplement 1.** Positions of *CrLFY* RNAi target sequences.
DOI: https://doi.org/10.7554/eLife.39625.018
**Figure supplement 2.** Generalized schematic of *CrLFY* RNAi constructs.
DOI: https://doi.org/10.7554/eLife.39625.019
**Figure supplement 3.** Gel blot analysis of *ZmUbi$_{pro}$::CrLFY1-i3* T$_1$ transgenic lines.
DOI: https://doi.org/10.7554/eLife.39625.020
**Figure supplement 4.** Binding site of *CrLFY* RNAi genotyping PCR primers.
DOI: https://doi.org/10.7554/eLife.39625.021
**Figure supplement 5.** Genotyping PCR confirms the presence of *CrLFY* RNAi T-DNA in transgenic lines and the absence of the RNAi hairpin in no hairpin control lines.
DOI: https://doi.org/10.7554/eLife.39625.022

which contained the complete transgene cassette and three lines that contained a fragment of the transgene which included the antibiotic resistance marker but not the RNAi hairpin (*Table 1*).

In the no hairpin control (NHC) plants, post-embryonic shoot development initiated with the production of simple, spade-like fronds from the shoot apex (*Figure 6A*) as in wild type. In eight transgenic lines, sub-populations of sporophytes developed in which this early stage of sporophyte development was perturbed, one line (E8) failing to initiate recognizable embryos (*Figure 6B*) and the remainder exhibiting premature shoot apex termination, typically after producing several distorted fronds (*Figure 6C–H*). Sub-populations of phenotypically normal transgenic sporophytes were also identified in some of these lines (*Figure 6I–L*). The two remaining lines exhibited less severe shoot phenotypes, one undergoing shoot termination after the production of simple (B19a) or lobed (B19b) fronds at the stage when control sporophytes produced complex dissected fronds (*Figure 6M–O*), and the other (D4b) completing sporophyte development but reduced in size to approximately 50% of controls (*Figure 6P,Q*). Despite the predicted sequence specificity of *CrLFY1-i3* and *CrLFY2-i4* (*Supplementary file 5*), qRT-PCR analysis found that all four RNAi constructs led to suppressed transcript levels of both *CrLFY* genes (*Figure 6R,S*). The severity of the shoot phenotype was correlated with the level of endogenous *CrLFY* transcripts detected across all lines (*Figure 6R, S*), with relative levels of both *CrLFY1* and *CrLFY2* significantly reduced compared to controls in all early-terminating sporophytes (E8, G13, C3, D2, D4a, D13, F9) ($p<0.01$ or less). In phenotypically normal transgenic siblings *CrLFY2* transcript levels were not significantly lower than controls (indeed in line D2, levels were higher $p<0.01$) whereas *CrLFY1* levels were significantly reduced ($p<0.0001$), as in arrested siblings. Together, these data indicate that *CrLFY2* can compensate for some loss of *CrLFY1*, but at least 22% of *CrLFY1* activity is required for normal development (line D4a pn, *Figure 6J,S*). It can thus be concluded that *CrLFY1* and *CrLFY2* act partially redundantly to maintain indeterminacy of the shoot apex in Ceratopteris, a role not found in the early divergent bryophyte *P. patens*, nor known to be retained in the majority of later diverging flowering plants.

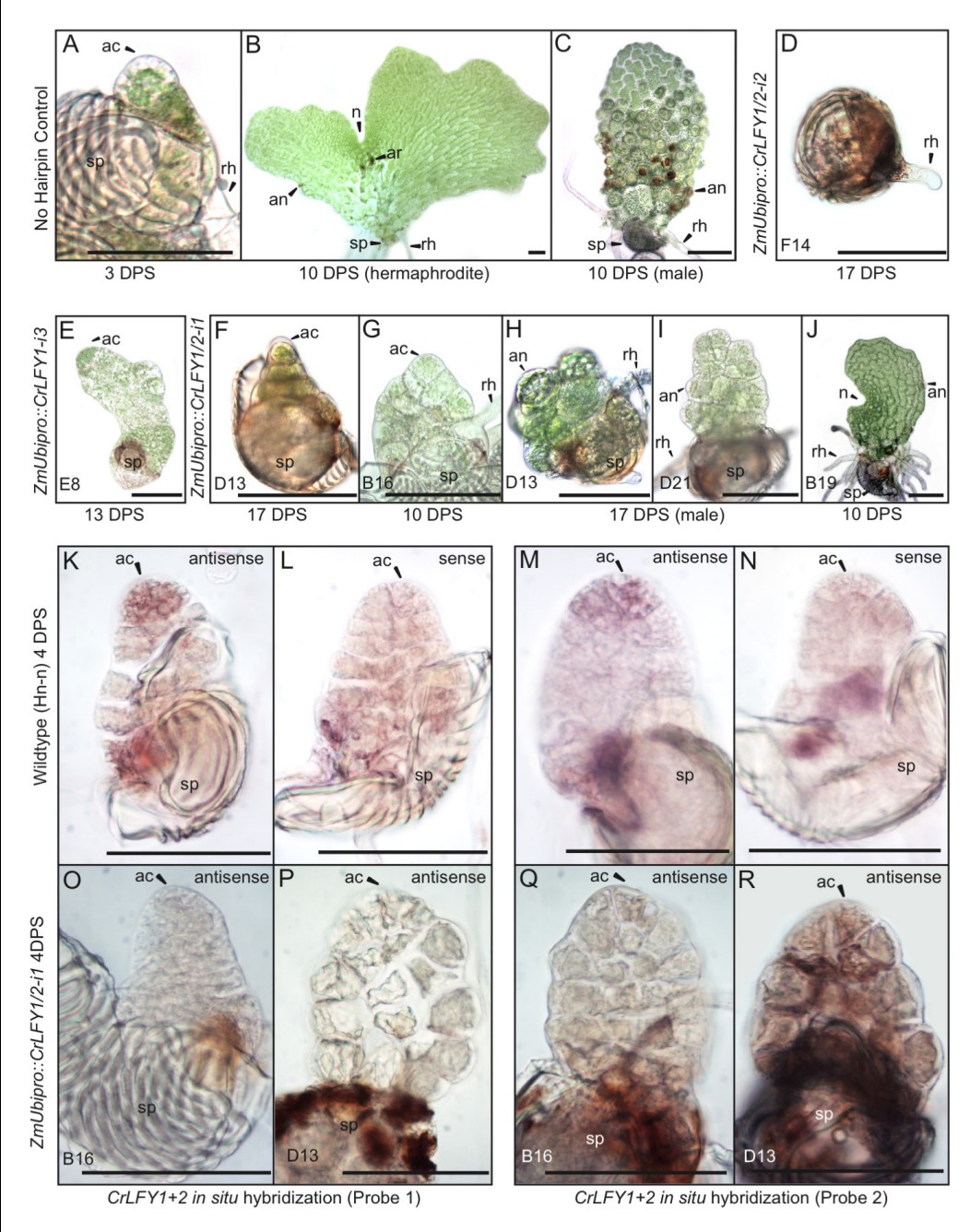

**Figure 7.** Suppression of *CrLFY* expression causes early termination of the Ceratopteris gametophyte apical cell. (A–C) In no hairpin control lines, the gametophyte established a triangular apical cell (ac) shortly after spore (sp) germination (A). Divisions of the apical cell established a photosynthetic thallus in both hermaphrodite and male gametophytes. At 10 days post spore sowing (DPS) both gametophyte sexes were approaching maturity, with the hermaphrodite (B) having formed a chordate shape from divisions at a lateral notch meristem (n) and having produced egg-containing archegonia (ar), sperm-containing antheridia (an), and rhizoids (rh). The male (C) had a more uniform shape with antheridia across the surface. These phenotypes were identical to wild-type. (D–J) When screened at 10–17 DPS, gametophytes from multiple RNAi lines (as indicated) exhibited developmental arrest, mostly associated with a failure of apical cell activity. Arrest occurred at various stages of development from failure to specify an apical cell, resulting in only a rhizoid being produced and no thallus (D) through subsequent thallus proliferation (E–I). Gametophyte development in one line progressed to initiation of the notch meristem but overall thallus size was severely reduced compared to wild-type (J). (K–R) In situ hybridization with antisense probes detected *CrLFY* transcripts in the apical cell and immediate daughter cells of wild-type gametophytes at 4

*Figure 7 continued on next page*

*Figure 7 continued*

DPS (**K, M**). No corresponding signal was detected in controls hybridized with sense probes (**L, N**). In the arrested gametophytes of two *ZmUbi_{pro}::CrLFY1/2-i1* lines *CrLFY* transcripts could not be detected (**O–R**), and transgene presence was confirmed (*Figure 7—figure supplement 1*). Scale bars = 100 μm.

DOI: https://doi.org/10.7554/eLife.39625.024

The following figure supplement is available for figure 7:

**Figure supplement 1.** Gametophytes exhibiting developmental arrest were transgenic.

DOI: https://doi.org/10.7554/eLife.39625.025

## *CrLFY* promotes apical cell divisions in the gametophyte

In six of the RNAi lines that exhibited sporophyte developmental defects, it was notable that 50–99% of gametophytes arrested development prior to the sporophyte phase of the lifecycle (*Table 1*). This observation suggested that LFY plays a role in Ceratopteris gametophyte development, a function not previously demonstrated in either bryophytes or angiosperms. During wild-type development, the Ceratopteris gametophyte germinates from a single-celled haploid spore, establishing a single apical cell (AC) within the first few cell divisions (*Figure 7A*). Divisions of the AC go on to form a two-dimensional photosynthetic thallus in both the hermaphrodite, where a notch meristem takes on growth (*Figure 7B*), and male sexes (*Figure 7C*) (*Banks, 1999*). In contrast, the gametophytes from six RNAi lines (carrying either *ZmUbi_{pro}::CrLFY1-i3*, *ZmUbi_{pro}::CrLFY1/2-i1* or *ZmUbi_{pro}:: CrLFY1/2-i2*) exhibited developmental arrest (*Figure 7D–J*), which in five lines clearly related to a failure of AC activity. The point at which AC arrest occurred varied, in the most severe line occurring prior to or during AC specification (*Figure 7D*) and in others during AC-driven thallus proliferation (*Figure 7E–I*). Failure of AC activity was observed in both hermaphrodites (*Figure 7E*) and males (*Figure 7H,I*). The phenotypically least-severe line exhibited hermaphrodite developmental arrest only after AC activity had been replaced by the notch meristem (*Figure 7J*). A role for *CrLFY* in maintenance of gametophyte AC activity was supported by the detection of *CrLFY* transcripts in the AC and immediate daughter cells of wild-type gametophytes by in situ hybridization (*Figure 7K–N*). By contrast *CrLFY* transcripts were not detected in arrested *ZmUbi_{pro}::CrLFY1/2-i1* lines (*Figure 7O– R*) in which the presence of the transgene was confirmed by genotyping of individual arrested gametophytes (*Figure 7—figure supplement 1*). *CrLFY1* and *CrLFY2* transcripts could not be clearly distinguished in situ due to sequence similarity (see *Supplementary file 6*), and hence the observed phenotypes could not be ascribed to a specific gene copy. However, these data support a role for at least one *CrLFY* homolog in AC maintenance during gametophyte development, and thus invoke a role for LFY in the regulation of apical activity in both the sporophyte and gametophyte phases of vascular plant development.

## Discussion

The results reported here reveal a role for LFY in the maintenance of apical cell activity throughout gametophyte and sporophyte shoot development in Ceratopteris. During sporophyte development, qRT-PCR and transgenic reporter lines demonstrated that *CrLFY1* is preferentially expressed in the shoot apex (whether formed during embryogenesis or de novo on fronds, and both before and after the reproductive transition); in emerging lateral organ (frond) primordia; and in pinnae and pinnules as they form on dissected fronds (*Figures 3–5*). Notably, active cell division is the main feature in all of these contexts. *CrLFY2* transcript levels were more uniform throughout sporophyte shoot development, in both dividing tissues and expanded fronds (*Figure 3*), and expression has previously been reported in roots (*Himi et al., 2001*). Simultaneous suppression of *CrLFY1* and *CrLFY2* activity by RNAi resulted in developmental arrest of both gametophyte and sporophyte shoot apices, with any fronds produced before termination of the sporophyte apex exhibiting abnormal morphologies (*Figures 6* and *7*). The severity of phenotypic perturbations in sporophytes of transgenic lines correlated with combined *CrLFY1* and *CrLFY2* transcript levels, with wild-type levels of *CrLFY2* able to fully compensate for up to a 70% reduction in *CrLFY1* levels (*Figure 6*). The duplicate *CrLFY* genes therefore act at least partially redundantly during shoot development in Ceratopteris.

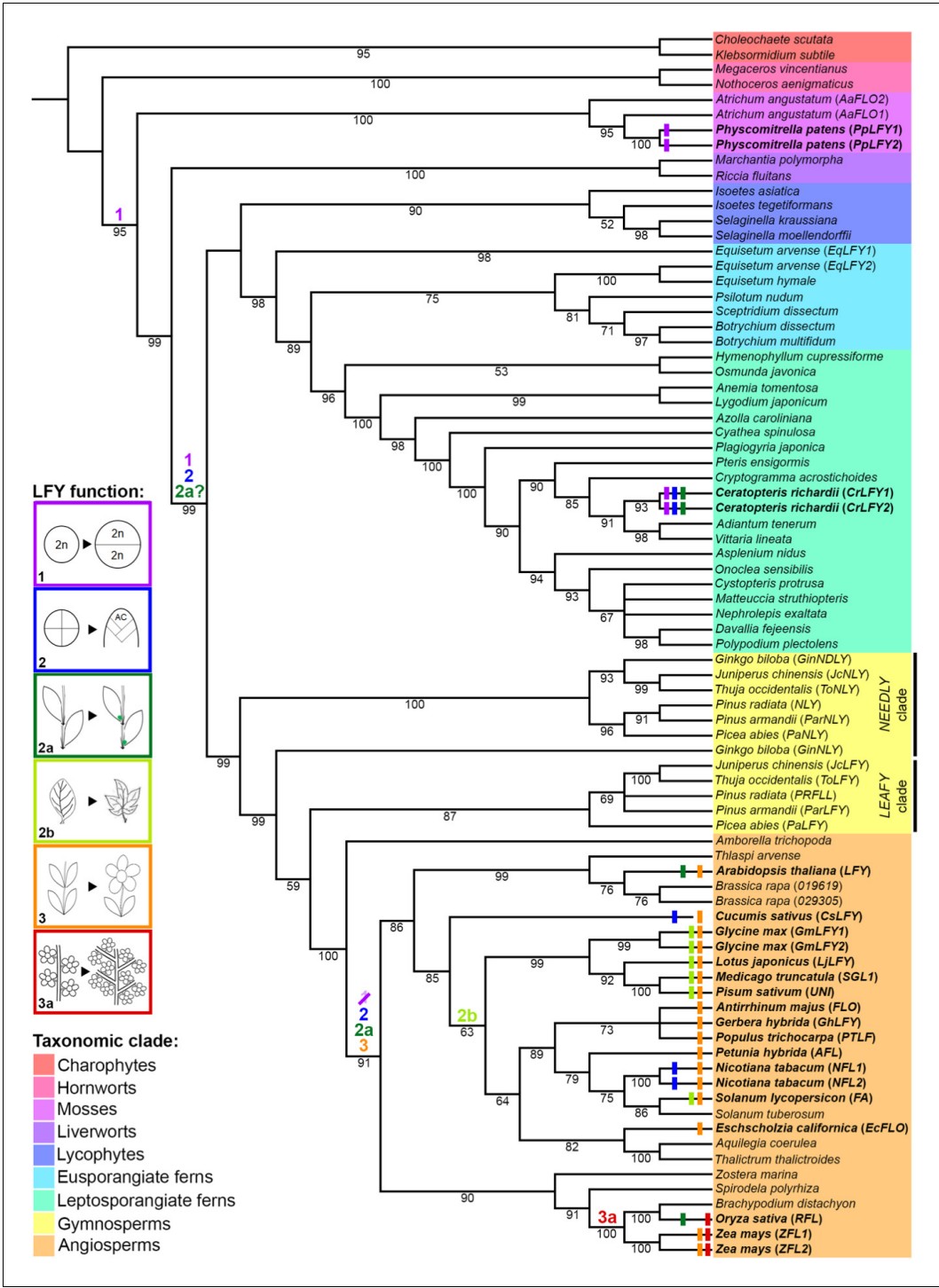

**Figure 8.** Evolutionary trajectory of LFY function. The phylogeny was reconstructed from selected LFY protein sequences representing all extant embryophyte lineages (as highlighted) and the algal sister-group. Coloured bars at the terminal branches represent different developmental functions of LFY determined from functional analysis in those species (see **Supplementary file 8** for references). Coloured numbers indicate the putative points of origin of different functions inferred from available data points across the tree. 1, cell division within the sporophyte zygote; 2, maintenance of indeterminate cell fate in vegetative shoots through proliferation of one or more apical cells (AC); 2a, maintenance of indeterminate cell fate in vegetative lateral/axillary apices; 2b, maintenance of indeterminate cell fate in the margins of developing lateral organs (compound leaves); 3, specification of floral meristem identity (determinate shoot development producing modified lateral organs) and shoot transition to the

*Figure 8 continued on next page*

*Figure 8 continued*

reproductive phase; 3a, maintenance of indeterminate cell fate in inflorescence lateral/branch meristems (in place of floral meristem fate).

DOI: https://doi.org/10.7554/eLife.39625.026

A function for LFY in gametophyte development has not previously been reported in any land plant species. In the moss *P. patens*, *PpLFY1* and *PpLFY2* are expressed in both the main and lateral apices of gametophytic leafy shoots but double loss-offunction mutants develop normally, indicating that LFY is not necessary for maintenance of apical cell activity in the gametophyte (*Tanahashi et al., 2005*). By contrast, loss of *CrLFY* expression from the gametophyte shoot apex results in loss of apical cell activity during thallus formation in Ceratopteris (*Figure 7*). The different DNA binding site preferences (and hence downstream target sequences) of PpLFY and CrLFY (*Sayou et al., 2014*) may be sufficient to explain the functional distinction in moss and fern gametophytes, but the conserved expression pattern is intriguing given that there should be no pressure to retain that pattern in *P. patens* in the absence of functional necessity. The thalloid gametophytes of the two other extant bryophyte lineages (liverworts and hornworts) resemble the fern gametophyte more closely than mosses (*Ligrone et al., 2012*), but LFY function in these contexts is not yet known. Overall the data are consistent with the hypothesis that in the last common ancestor of ferns and angiosperms, LFY functioned to promote cell proliferation in the thalloid gametophyte, a role that has been lost in angiosperms where gametophytes have no apical cell and are instead just few-celled determinate structures.

The range of reported roles for LFY in sporophyte development can be rationalized by hypothesizing three sequential changes in gene function during land plant evolution (*Figure 8*). First, the ancestral LFY function to promote early cell divisions in the embryo was retained in vascular plants after they diverged from the bryophytes, leading to conserved roles in *P. patens* (*Tanahashi et al., 2005*) and Ceratopteris (*Figure 4*). Second, within the vascular plants (preceding divergence of the ferns) this proliferative role expanded to maintain post-embryonic apical cell activity, and hence to enable indeterminate shoot growth. This is evidenced by *CrLFY* activity at the tips of shoots, fronds and pinnae (*Figures 4–6*), all of which develop from one or more apical cells (*Hill, 2001*; *Hou and Hill, 2004*). Whether fern fronds are homologous to shoots or to leaves in angiosperms is an area of debate (*Tomescu, 2009*; *Vasco et al., 2013*; *Harrison and Morris, 2018*), but there are angiosperm examples of LFY function in the vegetative SAM (*Ahearn et al., 2001*; *Zhao et al., 2018*), axillary meristems (*Kanrar et al., 2008*; *Rao et al., 2008*; *Chahtane et al., 2013*) and in actively dividing regions of compound leaves (*Hofer et al., 1997*; *Molinero-Rosales et al., 1999*; *Champagne et al., 2007*; *Wang et al., 2008*; *Monniaux et al., 2017*) indicating that a proliferative role in vegetative tissues has been retained in at least some angiosperm species. Consistent with the suggestion that the angiosperm floral meristem represents a modified vegetative meristem (*Theißen et al., 2016*), the third stage of LFY evolution could have been co-option and adaptation of this proliferation-promoting network into floral meristems, with subsequent restriction to just the flowering role in many species. This is consistent with multiple observations of *LFY* expression in both vegetative and reproductive shoots (developing cones) in gymnosperms (*Mellerowicz et al., 1998*; *Mouradov et al., 1998*; *Shindo et al., 2001*; *Carlsbecker et al., 2004*; *Vázquez-Lobo et al., 2007*; *Carlsbecker et al., 2013*; *Moyroud et al., 2017*) and suggests that pre-existing *LFY*-dependent vegetative gene networks might have been co-opted during the origin of specialized sporophyte reproductive axes in ancestral seed plants, prior to the divergence of angiosperms.

The proposed evolutionary trajectory for LFY function bears some resemblance to that seen for KNOX protein function. Class I *KNOX* genes are key regulators of indeterminacy in the vegetative shoot apical meristem of angiosperms (*Gaillochet et al., 2015*), and are required for compound leaf formation in both tomato and *Cardamine hirsuta* (*Bar and Ori, 2015*). In ferns, *KNOX* gene expression is observed both in the shoot apex and developing fronds (*Sano et al., 2005*; *Ambrose and Vasco, 2016*), and in *P. patens* the genes regulate cell division patterns in the determinate sporophyte (*Sakakibara et al., 2008*). It can thus be speculated that *LFY* and *KNOX* had overlapping functions in the sporophyte of the last common ancestor of land plants, but by the divergence of

ancestral angiosperms from gymnosperms, *KNOX* genes had come to dominate in vegetative meristems whereas *LFY* became increasingly specialized for floral meristem function. Unlike *LFY*, however, there is not yet any evidence for *KNOX* function in the gametophyte of any land plant lineage, and thus if a pathway for regulating stem cell activity was co-opted from the gametophyte into the sporophyte, the LFY pathway is the more likely one.

# Materials and methods

## Key resources table

| Reagent type (species) or resource | Designation | Source or reference | Identifiers | Additional information |
|---|---|---|---|---|
| Gene (*Ceratopteris richardii*) | *CrLEAFY1* (*CrLFY1*) | *Himi et al. (2001)*, PMID:11675598; This paper | NCBI:AB049974.2; NCBI:MH841970 | cDNA only; ORF plus contiguous promoter |
| Gene (*C. richardii*) | *CrLEAFY2* (*CrLFY2*) | *Himi et al. (2001)*, PMID:11675598; This paper | NCBI:AB049975.2; NCBI:MH841971 | cDNA only; ORF plus contiguous promoter |
| Strain, strain background (*C. richardii*) | Wild type (Hn-n) | *Warne and Hickok, 1987*, PMID:16665325 | | |
| Genetic reagent (*C. richardii*) | *CrLFY1/2-i1* | This paper | | *C. richardii* transgenic line; RNAi knockdown of *CrLFY1* and *CrLFY2* expression. |
| Genetic reagent (*C. richardii*) | *CrLFY1/2-i2* | This paper | | *C. richardii* transgenic line; RNAi knockdown of *CrLFY1* and *CrLFY2* expression. |
| Genetic reagent (*C. richardii*) | *CrLFY1-i3* | This paper | | *C. richardii* transgenic line; RNAi knockdown of *CrLFY1* expression. |
| Genetic reagent (*C. richardii*) | *CrLFY2-i4* | This paper | | *C. richardii* transgenic line; RNAi knockdown of *CrLFY2* expression. |
| Genetic reagent (*C. richardii*) | $CrLFY1_{pro}$::*GUS* | This paper | | C. richardii transgenic line; $CrLFY1_{pro}$::*GUS* reporter. |
| Genetic reagent (*C. richardii*) | $35S_{pro}$::*GUS* | This paper | | C. richardii transgenic line; $35S_{pro}$::*GUS* reporter. |
| Recombinant DNA reagent | $CrLFY1_{pro}$ | This paper | NCBI:MH841970 | CrLFY1 5′ genomic fragment; *Figure 1—figure supplement 2* |
| Recombinant DNA reagent | $CrLFY2_{pro}$ fragment 1 | This paper | NCBI:MH841971 | CrLFY2 5′ genomic fragment; *Figure 1—figure supplement 2* |
| Recombinant DNA reagent | $CrLFY2_{pro}$ fragment 2 | This paper | | CrLFY2 5′ genomic fragment; *Figure 1—figure supplement 2*; *Supplementary file 7* |
| Recombinant DNA reagent | GUS | *Ulmasov, 1997*, PMID:9401121 | | B-Glucuronidase (GUS) coding sequence |
| Recombinant DNA reagent | pANDA (RNAi vector) | *Miki and Shimamoto (2004)*, PMID:15111724 | | |
| Recombinant DNA reagent | pCR4-TOPO (Cloning vector) | Invitrogen | Thermo Scientific: K457502 | |
| Recombinant DNA reagent | pDONR207 (Gateway vector) | Invitrogen | | |
| Recombinant DNA reagent | pBOMBER (Binary vector) | *Plackett et al. (2015)*, PMID:26146510 | NCBI:MH841969 | Modified pART27 (PMID:1463857); Hygromycin resistance antibiotic selection marker |
| Recombinant DNA reagent | pART7 (Cloning vector) | Gleave 1992, PMID:1463857 | | |

*Continued on next page*

*Continued*

| Reagent type (species) or resource | Designation | Source or reference | Identifiers | Additional information |
|---|---|---|---|---|
| Recombinant DNA reagent | *ZmUbi_{pro}::CrLFY1/2-i1-pANDA* | This paper | | RNAi construct targeting *CrLFY1* and *CrLFY2*; *Figure 6—figure supplement 1*; *Figure 6—figure supplement 2* |
| Recombinant DNA reagent | *ZmUbi_{pro}::CrLFY1/2-i2-pANDA* | This paper | | RNAi construct targeting *CrLFY1* and *CrLFY2*; *Figure 6—figure supplement 1*; *Figure 6—figure supplement 2* |
| Recombinant DNA reagent | *ZmUbi_{pro}::CrLFY1-i3-pANDA* | This paper | | RNAi construct targeting *CrLFY1*; *Figure 6—figure supplement 1*; *Figure 6—figure supplement 2* |
| Recombinant DNA reagent | *ZmUbi_{pro}::CrLFY2-i4-pANDA* | This paper | | RNAi construct targeting CrLFY2; *Figure 6—figure supplement 1*; *Figure 6—figure supplement 2* |
| Recombinant DNA reagent | *CrLFY1_{pro}::GUS-pBOMBER* | This paper | | GUS reporter construct, *CrLFY1*; *Figure 4—figure supplement 1* |
| Recombinant DNA reagent | *35Spro::GUS-pBOMBER* | This paper | | GUS reporter construct, *35S* control; *Figure 4—figure supplement 1* |
| Recombinant DNA reagent | *CrLFY1 in situ* probe (antisense) | This paper | | In situ hybridisation probe; *Supplementary file 6* |
| Recombinant DNA reagent | *CrLFY1 in situ* probe (sense) | This paper | | In situ hybridisation probe; *Supplementary file 6* |
| Recombinant DNA reagent | *CrLFY2 in situ* probe (antisense) | This paper | | In situ hybridisation probe; *Supplementary file 6* |
| Recombinant DNA reagent | *CrLFY2 in situ* probe (sense) | This paper | | In situ hybridisation probe; *Supplementary file 6* |
| Recombinant DNA reagent | $^{32}$P-CrLFY1 probe 1 | This paper | | DNA gel blot probe for CrLFY1; *Figure 1—figure supplement 2* |
| Recombinant DNA reagent | $^{32}$P-CrLFY1 probe 2 | This paper | | DNA gel blot probe for CrLFY1; *Figure 1—figure supplement 2* |
| Recombinant DNA reagent | $^{32}$P-CrLFY2 probe 1 | This paper | | DNA gel blot probe for CrLFY2; *Figure 1—figure supplement 2* |
| Recombinant DNA reagent | $^{32}$P-CrLFY2 probe 1 | This paper | | DNA gel blot probe for CrLFY2; *Figure 1—figure supplement 2* |
| Recombinant DNA reagent | $^{32}$P-HygR probe | *Plackett et al. (2014)*, PMID:24623851 | | DNA gel blot probe, T-DNA specific; *Figure 4—figure supplement 1* |
| Recombinant DNA reagent | $^{32}$P-GUS probe | *Plackett et al. (2014)*, PMID:24623851 | | DNA gel blot probe, T-DNA specific; *Figure 4—figure supplement 1* |
| Recombinant DNA reagent | $^{32}$P-GUSlinker probe | This paper | | DNA gel blot probe, T-DNA specific; *Figure 6—figure supplement 2* |
| Sequence-based reagent | CrLFY1ampF | This paper | | ORF amplification, CrLFY1: 5'-ATGGATGTCTCTTTATTGCCAC-3' |
| Sequence-based reagent | CrLFY1ampR | This paper | | ORF amplification, CrLFY1: 5'-TCAATCATAGATGCAGCTATCACTG-3' |
| Sequence-based reagent | CrLFY1ampF | This paper | | ORF amplification, CrLFY2: 5'-ATGTTCCGATGGGAACAAAG-3' |
| Sequence-based reagent | CrLFY1ampR | This paper | | ORF amplification, CrLFY2: 5'-TTATTCATAGCTGCAGCTGTC-3' |
| Sequence-based reagent | CrLFY1invF | This paper | | Inverse PCR, CrLFY1: 5'-CTATGGAGTACGAAGCACCAC-3' |

*Continued on next page*

*Continued*

| Reagent type (species) or resource | Designation | Source or reference | Identifiers | Additional information |
|---|---|---|---|---|
| Sequence-based reagent | CrLFY1invF2 | This paper | | Inverse PCR, CrLFY1: 5'-CGATCATTTCTTGTACTGCTCTC-3' |
| Sequence-based reagent | CrLFY1invF3 | This paper | | Inverse PCR, CrLFY1: 5'-CAGTGCATGACCTTCGATATTG-3' |
| Sequence-based reagent | CrLFY1invR | This paper | | Inverse PCR, CrLFY1: 5'-CAGTTGTTTCGGATCTGCAG-3' |
| Sequence-based reagent | CrLFY1invR2 | This paper | | Inverse PCR, CrLFY1: 5'-CTCCGCTTTTCATTTGAGAACG-3' |
| Sequence-based reagent | CrLFY1invR3 | This paper | | Inverse PCR, CrLFY1: 5'-CAAGAACCGCTGGAGTAAAC-3' |
| Sequence-based reagent | CrLFY2invF | This paper | | Inverse PCR, CrLFY2: 5'-CTATGGTGTACGGAGCACTAC-3' |
| Sequence-based reagent | CrLFY2invF2 | This paper | | Inverse PCR, CrLFY2: 5'-CGTATCCAAAACAGCTTAAACTCC-3' |
| Sequence-based reagent | CrLFY2invF3 | This paper | | Inverse PCR, CrLFY2: 5'-CACTAAAGGTGCTGCTATCAAC-3' |
| Sequence-based reagent | CrLFY2invF4 | This paper | | Inverse PCR, CrLFY2: 5'-CATTGTGCTGACCTTGTGAAG-3' |
| Sequence-based reagent | CrLFY2invF5 | This paper | | Inverse PCR, CrLFY2: 5'-CGCAAAGGTTGGAAAAGAGAAC-3' |
| Sequence-based reagent | CrLFY2invF6 | This paper | | Inverse PCR, CrLFY2: 5'-CGACAACGGATCATAACCATC-3' |
| Sequence-based reagent | CrLFY2 invF7 | This paper | | Inverse PCR, CrLFY2: 5'-CAATAGTAGATTCTCCCTCCTTTAC-3' |
| Sequence-based reagent | CrLFY2invF8 | This paper | | Inverse PCR, CrLFY2: 5'-GCTCTTTAATTTGAATCACGTGTG-3' |
| Sequence-based reagent | CrLFY2invF9 | This paper | | Inverse PCR, CrLFY2: 5'-GAACAATGTGCATGCGACTC-3' |
| Sequence-based reagent | CrLFY2invF10 | This paper | | Inverse PCR, CrLFY2: 5'-CATGTTCCGATGGGAACAAAG-3' |
| Sequence-based reagent | CrLFY2invF11 | This paper | | Inverse PCR, CrLFY2: 5'-CATAGGGAACTCTGTAATGATGC-3' |
| Sequence-based reagent | CrLFY2invF12 | This paper | | Inverse PCR, CrLFY2: 5'-GTTTCCAGATACTGCTGCTC-3' |
| Sequence-based reagent | CrLFY2invF13 | This paper | | Inverse PCR, CrLFY2: 5'-CATAGATGATGCCAGTATACTCC-3' |
| Sequence-based reagent | CrLFY2invF14 | This paper | | Inverse PCR, CrLFY2: 5'-GCTCACTATCCACAATTCATACAC-3' |
| Sequence-based reagent | CrLFY2invF15 | This paper | | Inverse PCR, CrLFY2: 5'-GTTCGTATCTGATACTTGTTTCGTG-3' |
| Sequence-based reagent | CrLFY2invF16 | This paper | | Inverse PCR, CrLFY2: 5'-CTTACTCCACGAATGCATGC-3' |

Continued

| Reagent type (species) or resource | Designation | Source or reference | Identifiers | Additional information |
|---|---|---|---|---|
| Sequence-based reagent | CrLFY2invR | This paper | | Inverse PCR, CrLFY2: 5'-CAGTTGTCAC AGAGGTAGCAG-3' |
| Sequence-based reagent | CrLFY2invR2 | This paper | | Inverse PCR, CrLFY2: 5'-CCTTACGATG TATTACCCTTTGTTC-3' |
| Sequence-based reagent | CrLFY2invR3 | This paper | | Inverse PCR, CrLFY2: 5'-CAGTGACTA GGATGTCTGATACAG-3' |
| Sequence-based reagent | CrLFY2invR4 | This paper | | Inverse PCR, CrLFY2: 5'-GAAGGAGCT GAAAATGCAACTC-3' |
| Sequence-based reagent | CrLFY2invR5 | This paper | | Inverse PCR, CrLFY2: 5'-CCTGCCTCC TATGAAAACAC-3' |
| Sequence-based reagent | CrLFY2invR6 | This paper | | Inverse PCR, CrLFY2: 5'-CCTGTAAAGG AGGGAGAATCTAC-3' |
| Sequence-based reagent | CrLFY2invR7 | This paper | | Inverse PCR, CrLFY2: 5'-GCACTCCAAC GATGATGATAC-3' |
| Sequence-based reagent | CrLFY2invR8 | This paper | | Inverse PCR, CrLFY2: 5'-GCTGTACTA AGGCATCAATTCAG-3' |
| Sequence-based reagent | CrLFY2invR9 | This paper | | Inverse PCR, CrLFY2: 5'-CATCTATGATA GCACAACATCACTC-3' |
| Sequence-based reagent | CrLFY2invR10 | This paper | | Inverse PCR, CrLFY2: 5'-CACAACATC ACTCAGGACTC-3' |
| Sequence-based reagent | CrLFY2invR11 | This paper | | Inverse PCR, CrLFY2: 5'-CTGCCTCCTA TGAAAACACAAG-3' |
| Sequence-based reagent | CrLFY2invR12 | This paper | | Inverse PCR, CrLFY2: 5'-CTAGTCTTTG ATGAGGTTTCATGTC-3' |
| Sequence-based reagent | CrLFY2invR13 | This paper | | Inverse PCR, CrLFY2: 5'-CATGCAAGA AGCATGCAATTC-3' |
| Sequence-based reagent | CrLFY2invR14 | This paper | | Inverse PCR, CrLFY2: 5'-GTGTCTCCA GTAAGTATGAAACAAG-3' |
| Sequence-based reagent | CrLFY2invR15 | This paper | | Inverse PCR, CrLFY2: 5'-CATGAGGCC GTCAGACTTAC-3' |
| Sequence-based reagent | CrLFY2invR16 | This paper | | Inverse PCR, CrLFY2: 5'-CGTAACAGA CGAGCTCGATATAATAG-3' |
| Sequence-based reagent | CrLFY2invR17 | This paper | | Inverse PCR, CrLFY2: 5'-CTCTTTGCTCA TATAGCTTCAAGC-3' |
| Sequence-based reagent | CrLFY1 + 2 (1)-RNAi-F | This paper | | T-DNA cloning, CrLFY1/2-i1: 5'-ATGGGT TTCACTGTGAATAC-3' |
| Sequence-based reagent | CrLFY1 + 2 (1)-RNAi-R | This paper | | T-DNA cloning, CrLFY1/2-i1: 5'-TCTCCTC TTTGTTCCCTTGTG-3' |

*Continued on next page*

*Continued*

| Reagent type (species) or resource | Designation | Source or reference | Identifiers | Additional information |
|---|---|---|---|---|
| Sequence-based reagent | CrLFY1 + 2 (2)-RNAi-F | This paper | | T-DNA cloning, CrLFY1/2-i2: 5'-ATGGG TTTCACTGTTAGTAC-3' |
| Sequence-based reagent | CrLFY1 + 2 (2)-RNAi-R | This paper | | T-DNA cloning, CrLFY1/2-i2: 5'-TCTCCT CTTTGTTCCCTGGTG-3' |
| Sequence-based reagent | CrLFY1-RNAi-F | This paper | | T-DNA cloning, CrLFY1-i3: 5'-CCTTTTCT TGCTAATGATGGC-3' |
| Sequence-based reagent | CrLFY1-RNAi-R | This paper | | T-DNA cloning, CrLFY1-i3: 5'-CAAACAAA CTTGAAAATGATAC-3' |
| Sequence-based reagent | CrLFY2-RNAi-F | This paper | | T-DNA cloning, CrLFY2-i4: 5'-GCCATTG CTAGCAAGGTTAT-3' |
| Sequence-based reagent | CrLFY2-RNAi-R | This paper | | T-DNA cloning, CrLFY2-i4: 5'-CACTGCT TTGAAACTAAAAC-3' |
| Sequence-based reagent | pCrLFY1amp-NotF | This paper | | T-DNA cloning, CrLFY1$_{pro}$: 5'-CAGCGGCCGCTTAGATGG CTTGAGATGCTAC-3' |
| Sequence-based reagent | pCrLFY1amp-XbaR | This paper | | T-DNA cloning, CrLFY1$_{pro}$: 5'-CATCTAGAG GAGGCACTTCTTTACGTG-3' |
| Sequence-based reagent | GUSamp-XbaF | This paper | | T-DNA cloning, GUS CDS: 5'-CATCTAGAC AATGGTAAGCTTAGCGGG-3' |
| Sequence-based reagent | GUSamp-XbaR | This paper | | T-DNA cloning, GUS CDS: 5'-CCATCTAGA TTCATTGTTTGCCTCCCTG-3' |
| Sequence-based reagent | qCrLFY1_F2 | This paper | | qRT-PCR, CrLFY1: 5'-GTCCGCT ATTCGTGCAGAGA-3' |
| Sequence-based reagent | qCrLFY1_R2 | This paper | | qRT-PCR, CrLFY1 : 5'-AATTCAAGGGGG CATTGGGT-3' |
| Sequence-based reagent | qCrLFY2_F3 | This paper | | qRT-PCR, CrLFY2: 5'-GCAGTGACAATGAAGGACGC-3' |
| Sequence-based reagent | qCrLFY2_R3 | This paper | | qRT-PCR, CrLFY2: 5'-AGAATCGTGCACACTGCTCA-3' |
| Sequence-based reagent | qCrTBPb_F | *Ganger et al. (2015)*, DOI: 10.1139/cjb-2014–0202 | | qRT-PCR, CrTBP: 5'-ATGAGCCAGAGCTTTTCCCC-3' |
| Sequence-based reagent | qCrTBPb_R | *Ganger et al. (2015)*, DOI: 10.1139/cjb-2014–0202 | | qRT-PCR, CrTBP: 5'-TTCGTCTCTGACCTTTGCCC-3' |
| Sequence-based reagent | qCrACT1_F | *Ganger et al. (2015)*, DOI: 10.1139/cjb-2014–0202 | | qRT-PCR, CrActin1: 5'-GAGAGAGGCTA CTCTTTCACAACC-3' |
| Sequence-based reagent | qCrACT1_R | *Ganger et al. (2015)*, DOI: 10.1139/cjb-2014–0202 | | qRT-PCR, CrActin1: 5'-AGGAAGTTCGTA ACTCTTCTCCAA-3' |
| Sequence-based reagent | CrLFY1_ISH_F | This paper | | In situ probes, CrLFY1: 5'-GAGGCATACA CACACGCAGT-3' |
| Sequence-based reagent | CrLFY1_ISH_R | This paper | | In situ probes, CrLFY1: 5'-TCAATCATAGAT GCAGCTATCACTG-3 |

*Continued on next page*

*Continued*

| Reagent type (species) or resource | Designation | Source or reference | Identifiers | Additional information |
|---|---|---|---|---|
| Sequence-based reagent | CrLFY2_ISH_F | This paper | | In situ probes, CrLFY2: 5'-GGCTGGTTGTTA CGGATAGC-3' |
| Sequence-based reagent | CrLFY2_ISH_R | This paper | | In situ probes, CrLFY2: 5'-TTATTCATAG CTGCAGCTGTCACTG-3' |
| Sequence-based reagent | CrLFY1_Probe1F | This paper | | Copy number analysis, CrLFY1 probe 1: 5'-CAGG CACAAGGGAACAAAG-3' |
| Sequence-based reagent | CrLFY1_Probe1R | This paper | | Copy number analysis, CrLFY1 probe 1: 5'-CA TAGATGCAGCTATCACTGTC-3' |
| Sequence-based reagent | CrLFY1_Probe2F | This paper | | Copy number analysis, CrLFY1 probe 2: 5'-CACTTGAAGGTAAGCT TTATTGTAAGG-3' |
| Sequence-based reagent | CrLFY1_Probe2R | This paper | | Copy number analysis, CrLFY1 probe 2: 5'-CAATA TTTCCGACTATACATTGAGGC-3' |
| Sequence-based reagent | CrLFY2_Probe1F | This paper | | Copy number analysis, CrLFY2 probe 1: 5'-CAGGCA CCAGGGAACAAAG-3' |
| Sequence-based reagent | CrLFY2_Probe1R | This paper | | Copy number analysis, CrLFY2 probe 1: 5'-CATAGC TGCAGCTGGTCACTGTC-3' |
| Sequence-based reagent | CrLFY2_Probe2F | This paper | | Copy number analysis, CrLFY2 probe 2: 5'-CTGTAG AAGGTAAGATTCTGCTC-3' |
| Sequence-based reagent | CrLFY2_Probe2R | This paper | | Copy number analysis, CrLFY2 probe 2: 5'-GCTT ATGGTACAGAATAAGTAGAGG-3' |
| Sequence-based reagent | HygF2 | *Plackett et al. (2014)*, PMID:24623851 | | T-DNA gel blot probe, Hyg$^R$: 5'-CTTCTACA CAGCCATCGGTC-3' |
| Sequence-based reagent | HygR | *Plackett et al. (2014)*, PMID:24623851 | | T-DNA gel blot probe, Hyg$^R$: 5'-CCGATGGT TTCTACAAAGATCG-3' |
| Sequence-based reagent | GH3seqF3 | *Plackett et al. (2014)*, PMID:24623851 | | T-DNA gel blot probe, GUS: 5'-CTTCGCT GTACAGTTCTTTCG-3' |
| Sequence-based reagent | GH3seqR4 | *Plackett et al. (2014)*, PMID:24623851 | | T-DNA gel blot probe, GUS: 5'-CACTCATT ACGGCAAAGTGTG-3' |
| Sequence-based reagent | GUSlinkerseqF | This paper | | T-DNA gel blot probe, RNAi: 5'-CTGATT AACCACAAACCGTTCTAC-3' |
| Sequence-based reagent | GUSlinkerseqR | This paper | | T-DNA gel blot probe, RNAi: 5'-CTGATA CTCTTCACTCCACATG-3' |
| Sequence-based reagent | HPT-F | *Miki and Shimamoto (2004)*, PMID:15111724 | | RNAi genotyping, Hyg$^R$: 5'-GAGCCTGACCTA TTGCATCTCC-3' |
| Sequence-based reagent | HPT-R | *Miki and Shimamoto (2004)*, PMID:15111724 | | RNAi genotyping, Hyg$^R$: 5'-GGCCTCCAG AAGAAGATGTTGG-3' |

*Continued on next page*

*Continued*

| Reagent type (species) or resource | Designation | Source or reference | Identifiers | Additional information |
|---|---|---|---|---|
| Sequence-based reagent | pVec8F | *Miki and Shimamoto (2004)*, PMID:15111724 | | RNAi genotyping, RNAi hairpin: 5'-TTTAGC CCTGCCTTCATACG-3' |
| Sequence-based reagent | pVec8R | *Miki and Shimamoto (2004)*, PMID:15111724 | | RNAi genotyping, RNAi hairpin: 5'-ATTGC CAAATGTTTGAACGA-3' |
| Sequence-based reagent | PW64F | This paper | | RNAi genotyping, RNAi hairpin: 5'-CATGAA GATGCGGACTTACG-3' |
| Sequence-based reagent | PW64R | This paper | | RNAi genotyping, RNAi hairpin: 5'-ATCCAC GCCGTATTCGG-3' |
| Sequence-based reagent | pCrLFY1genoF1 | This paper | | $CrLFY1_{pro}::GUS$ genotyping: 5'-CTTAGA TGGCTTGAGATGCTAC-3' |
| Sequence-based reagent | pCrLFY1genoF2 | This paper | | $CrLFY1_{pro}::GUS$ genotyping: 5'-CTCTCT TCTTGCTTGTGTTGTG-3' |
| Sequence-based reagent | pCrLFY1genoF3 | This paper | | $CrLFY1_{pro}::GUS$ genotyping: 5'-CAACTGGCAACAGGTGATG-3' |
| Sequence-based reagent | pCrLFY1genoF4 | This paper | | $CrLFY1_{pro}::GUS$ genotyping: 5'-CAGTCTTAGTTCAACTGCATTCG-3' |
| Sequence-based reagent | pCrLFY1genoR | This paper | | $CrLFY1_{pro}::GUS$ genotyping: 5'-AGGAGGCACTTCTTTACGTG-3' |
| Sequence-based reagent | GUSgenoR | This paper | | $CrLFY1_{pro}::GUS + 35S_{pro}::GUS$ genotyping: 5'-CATTGTTTG CCTCCCTGC-3' |
| Sequence-based reagent | 35SgenoF | This paper | | $35S_{pro}::GUS$ genotyping: 5'-CTGAGCTTAACAGCACAGTTG-3' |
| Sequence-based reagent | OCS3'genoR | This paper | | $35S_{pro}::GUS$ genotyping: 5'-CATCACTAGTAAGCTAGCTTGC-3' |
| Commercial assay or kit | Phusion high-fidelity polymerase | Thermo Scientific | Thermo Scientific: F530S | |
| Commercial assay or kit | Gateway LR clonase II enzyme mix | Invitrogen | Thermo Scientific: 11791100 | |
| Commercial assay or kit | QIAGEN Plasmid Maxi Kit | QIAGEN | QIAGEN:12163 | |
| Commercial assay or kit | Whatman Nytran nylon blotting membrane | GE Healthcare | GE Healthcare: 10416294 | |
| Commercial assay or kit | Random Primers DNA Labelling kit | Invitrogen | Thermo Scientific: 18187013 | |
| Commercial assay or kit | Carestream Kodak autoradiography GBX developer and fixer | Sigma-Aldrich | Sigma-Aldrich: Z354147 | |
| Commercial assay or kit | Carestream Kodak Biomax XAR film | Sigma-Aldrich | Sigma-Aldrich:F5763 | |
| Commercial assay or kit | iTaq universal SYBR Green mastermix | Bio-Rad | Bio-Rad:1725120 | |
| Commercial assay or kit | DIG-labelling mix | Roche Applied Sciences | Roche:11277073910 | |
| Commercial assay or kit | T3 RNA polymerase | Roche Applied Sciences | Roche:11031163001 | |
| Commercial assay or kit | T7 RNA polymerase | Roche Applied Sciences | Roche:10881767001 | |

*Continued on next page*

*Continued*

| Reagent type (species) or resource | Designation | Source or reference | Identifiers | Additional information |
|---|---|---|---|---|
| Commercial assay or kit | Anti-DIG antibody | Roche Applied Sciences | Roche:11093274910; RRID:AB_2313639 | |
| Commercial assay or kit | NBT/BCIP stock solution | Roche Applied Sciences | Roche:11681451001 | |
| Chemical compound, drug | Potassium ferricyanide ($K_3Fe(CN)_6$) | Sigma-Aldrich | Sigma:P8131 | |
| Chemical compound, drug | X-GlcA (CHA salt) | Melford Scientific | Melford:MB1021 | |
| Chemical compound, drug | CTP, [$\alpha$-$^{32}$P] | Perkin Elmer | Perkin Elmer: BLU008H250UC | |
| Software, algorithm | IQ-TREE | *Nguyen et al. (2015)*, PMID:25371430 | | http://www.iqtree.org/ |
| Software, algorithm | iTOL | *Letunic and Bork (2016)*, PMID:27095192 | | https://itol.embl.de/ |
| Software, algorithm | ClustalW | *Li et al. (2015)*, PMID:25845596 | RRID:SCR_002909 | https://www.ebi.ac.uk/Tools/msa/clustalw2/ |
| Software, algorithm | TBLASTX | *Altschul et al. (1990)*, PMID:2231712 | RRID:SCR_011823 | https://blast.ncbi.nlm.nih.gov/Blast.cgi |
| Software, algorithm | GeneWise | *Birney et al. (2004)*, PMID:15123596 | RRID:SCR_015054 | https://www.ebi.ac.uk/Tools/psa/genewise |
| Software, algorithm | GraphPad Prism | GraphPad Software Inc. | RRID:SCR_002798 | https://www.graphpad.com/scientific-software/prism/ |
| Software, algorithm | Adobe Photoshop CS4 | Adobe | RRID:SCR_014199 | |
| Other | Biolistic PDS-1000/He Particle Delivery System | Bio-Rad | Bio-Rad:1652257 | |
| Other | CFX Connect Real-Time PCR Detection System | Bio-Rad | Bio-Rad:1855201 | |
| Other | Zeiss Axioplan microscope | Zeiss | | |
| Other | Nikon Microphot-FX microscope | Nikon | | |
| Other | MicroPublisher 3.3 RTV camera | Qimaging | | |

## Plant materials and growth conditions

All experimental work was conducted using *Ceratopteris richardii* strain Hn-n (*Warne and Hickok, 1987*). Plant growth conditions for Ceratopteris transformation and DNA gel blot analysis of transgenic lines were as previously described (*Plackett et al., 2015*).

## Phylogenetic analysis

A dataset of 99 aligned LFY protein sequences from a broad range of streptophytes was first retrieved from *Sayou et al. (2014)*. The dataset was pruned and then supplemented with further sequences (*Supplementary file 1*) to enable trees to be inferred that would (i) provide a more balanced distribution across the major plant groups and (ii) infer fern relationships. Only a subset of available angiosperm sequences was retained (keeping both monocot and dicot representatives) but protein sequences from other angiosperm species where function has been defined through loss-of-function analyses were added from NCBI – *Antirrhinum majus* FLO AAA62574.1 (*Coen et al., 1990*), *Pisum sativum* UNI AAC49782.1 (*Hofer et al., 1997*), *Cucumis sativus* CsLFY XP_004138016.1 (*Zhao et al., 2018*), *Medicago truncatula* SGL1 AY928184 (*Wang et al., 2008*), *Petunia hybrida* ALF AAC49912.1 (*Souer et al., 1998*), *Nicotiana tabacum* NFL1 AAC48985.1 and NFL2 AAC48986.1 (*Kelly, 1995*), *Eschscholzia californica* EcFLO AAO49794.1 (*Busch and Gleissberg, 2003*), *Gerbera hybrida* cv. 'Terraregina' GhLFY ANS10152.1 (*Zhao et al., 2016*), *Lotus japonicus* LjLFY AAX13294.1 (*Dong et al., 2005*) and *Populus trichocarpa* PTLF AAB51533.1 (*Rottmann et al., 2000*). To provide

better resolution within and between angiosperm clades, sequences from *Spirodela polyrhiza* (32G0007500), *Zostera marina* (27g00160.1), *Aquilegia coerulea* (5G327800.1) and *Solanum tuberosum* (PGSC0003DMT400036749) were added from Phytozome v12.1 (https://phytozome.jgi.doe. gov/pz/portal.html). Genome sequence from the early-diverging Eudicot *Thalictrum thalictroides* was searched by TBLASTX (*Altschul et al., 1990*) (https://blast.ncbi.nlm.nih.gov/Blast.cgi?PRO-GRAM=tblastx&PAGE_TYPE=BlastSearch&BLAST_SPEC=&LINK_LOC=blasttab) with nucleotide sequence from the Arabidopsis *LFY* gene. A gene model was derived from sequence in two contigs (108877 and 116935) using Genewise (*Birney et al., 2004*) (https://www.ebi.ac.uk/Tools/psa/gene-wise/). Gymnosperm sequences were retained from *Ginkgo biloba* and from a subset of conifers included in *Sayou et al. (2014)*, whilst sequences from conifers where *in situ* hybridization patterns have been reported were added from NCBI – *Pinus radiata* PRFLL AAB51587.1 and NLY AAB68601.1 (*Mellerowicz et al., 1998*; *Mouradov et al., 1998*) and *Picea abies* PaLFY AAV49504.1 and PaNLY AAV49503.1 (*Carlsbecker et al., 2004*). Fern sequences were retained except *Angiopteris spp* sequences which consistently disrupted the topology of the tree by grouping with gymnosperms. To better resolve relationships within the ferns, additional sequences were identified in both NCBI and 1KP (*Matasci et al., 2014*) databases. The protein sequence from *Matteuccia struthiopteris* AAF77608.1 MatstFLO (*Himi et al., 2001*) was retrieved from NCBI. Further sequences from horsetails (2), plus eusporangiate (1) and leptosporangiate (53) ferns were retrieved from the 1KP database (https://db.cngb.org/blast/) using BLASTP and the MatstFLO sequence as a query. Lycophyte and bryophyte sequences were all retained, but the liverwort *Marchantia polymorpha* predicted ORF sequence was updated from Phytozome v12.1 (Mpo0113s0034.1.p), the hornwort *Nothoceros* genome scaffold was replaced with a translated full length cDNA sequence (AHJ90704.1) from NCBI and two additional lycophyte sequences were added from the 1KP dataset (*Isoetes tegetiformans* scaffold 2013584 and *Selaginella kraussiana* scaffold 2008343). All of the charophyte scaffold sequences were substituted with *Coleochaete scutata* (AHJ90705.1) and *Klebsormidium subtile* (AHJ90707.1) translated full-length cDNAs from NCBI.

The new/replacement sequences were trimmed and amino acids aligned to the existing alignment from *Sayou et al. (2014)* using CLUSTALW (*Li et al., 2015*) (*Supplementary file 2* and *3*). The best-fitting model parameters (JTT + I + G4) were estimated and consensus phylogenetic trees were run using Maximum Likelihood from 1000 bootstrap replicates, using IQTREE (*Nguyen et al., 2015*). Two trees were inferred. The first contained only a subset of fern and allied sequences to achieve a more balanced distribution across the major plant groups (81 sequences in total) (*Figure 8*), whereas the second used the entire dataset (120 sequences ~ 50% of which are fern and allied sequences – *Figure 1—figure supplement 1*). The data were imported into ITOL (*Letunic and Bork, 2016*) to generate the pictorial representations. All branches with less than 50% bootstrap support were collapsed. Relationships within the ferns (*Figure 1*) were represented by pruning the lycophyte and fern sequences (68 in total) from the tree containing all available fern sequences (*Figure 1—figure supplement 1*).

## *CrLFY* locus characterization and DNA gel blot analysis

Because no reference genome has yet been established for Ceratopteris (or any fern), *CrLFY* copy number was quantified by DNA gel blot analysis. Ceratopteris genomic DNA was hybridized using both the highly conserved LFY DNA-binding domain diagnostic of the *LFY* gene family (*Maizel et al., 2005*) and also gene copy-specific sequences (*Figure 1—figure supplement 2*). CrLFY1 and CrLFY2 share 85% amino acid similarity, compared to 65% and 44% similarity of each to AtLFY. DNA gel blotting and hybridization was performed as described previously (*Plackett et al., 2014*). The results supported the presence of only two copies of *LFY* within the Ceratopteris genome. All primers used in probe preparation are supplied in the Key Resources Table.

Genomic sequences for *CrLFY1* and *CrLFY2* open reading frames (ORFs) were amplified by PCR from wild-type genomic DNA using primers designed against published transcript sequences (*Himi et al., 2001*). ORFs of 1551 bp and 2108 bp were obtained, respectively (*Figure 1—figure supplement 2*). Exon structure was determined by comparison between genomic and transcript sequences. The native promoter region of *CrLFY1* was amplified from genomic template by sequential rounds of inverse PCR with initial primer pairs designed against published *CrLFY1* 5'UTR sequence and additional primers subsequently designed against additional contiguous sequence that was retrieved. A 3.9 kb contiguous promoter fragment was isolated for *CrLFY1* containing the

entire published 5'UTR and 1.9 kb of additional upstream sequence (*Figure 1—figure supplement 2*). Repeated attempts were made to obtain a *CrLFY2* promoter fragment but this proved impossible in the absence of a reference genome. Some sequence contiguous with the *CrLFY2* ORF was obtained by inverse PCR using primers designed against the previously published 5'UTR sequence of the *CrLFY2* transcript (*Himi et al., 2001*). This sequence was extended to 1016 bp in length using additional primers against the isolated genomic sequence but this fragment did not contain the entire published 5'UTR. Numerous rounds of inverse PCR generated a second 3619 bp genomic fragment containing sequence identical to the remaining 5'UTR (see *Figure 1—figure supplement 2*, *Supplementary file 7*) but the presumed connecting sequence between these two fragments could not be amplified despite many attempts. It was eventually concluded that either the intervening promoter fragment was too long to amplify or that it was too GC rich for amplification. All primers used in ORF amplification and inverse PCR are supplied in the Key Resources table. The contiguous sequences obtained for the *CrLFY1* and *CrLFY2* genomic loci have been submitted to Genbank (accessions MH841970 and MH841971, respectively).

## qRT-PCR analysis of gene expression

RNA was extracted from Ceratopteris tissues using the Spectrum Total Plant RNA kit (Sigma-Aldrich, St. Louis, MO) and 480 ng were used as template in iScript cDNA synthesis (Bio-Rad). *CrLFY1* and *CrLFY2* locus-specific qRT-PCR primers were designed spanning intron 1. Amplification specificity of primers was validated via PCR followed by sequencing. qRT-PCR of three biological replicates and three technical replicates each was performed in a Bio-Rad CFX Connect with iTaq Universal SYBR Green Supermix (Bio-Rad, Hercules, CA). Primer amplification efficiency was checked with a cDNA serial dilution. Efficiency was determined using the slope of the linear regression line as calculated by Bio-Rad CFX Connect software. Primer specificity was tested via melting curve analysis, resulting in a single peak per primer set. *CrLFY* expression was calculated using the $2^{-\Delta\Delta Ct}$ method (*Livak and Schmittgen, 2001*) and normalized against the geometric mean of the expression of two endogenous reference genes (*Hellemans et al., 2007*), *CrACTIN1* and *CrTATA-BINDING PROTEIN (TBP)* (*Ganger et al., 2015*). The standard deviation of the Ct values of each reference gene was calculated to ensure minimal variation (<3%) in gene expression. Error bars represent ± the standard error of the mean of the $2^{\Delta\Delta Ct}$ values. All primers used in qRT-PCR are supplied in the Key Resources table.

Relative expression values of *CrLFY* from qRT-PCR were compared by one or two-way analysis of variance (ANOVA) for developmental stages followed by Tukey's or Sidak's multiple comparisons, respectively. To test whether genes were downregulated in transgenic RNAi lines, two-way ANOVA was perfomed with gene (*CrLFY1* or *CrLFY2*) and transgenic line as factors, with 'gene' as a repeated factor when all transgenic lines had the same number of replicates. Where appropriate, expression of each gene in each line was compared to the expression of the respective control by Dunnet comparisons. Control plants had been transformed and were hygromycin-resistant, but did not contain the RNAi hairpin that triggers gene silencing (non-hairpin controls, NHC). For all experiments, NHCs were grown alongside transgenic lines. qRT-PCR of transgenic lines was necessarily conducted across several plates, each including a representative NHC, and statistical comparisons were performed within each plate relative to its respective control. The significance threshold (p) was set at 0.05. All statistical analyses were performed in Prism v. 6.0 (GraphPad Software, Inc., La Jolla, CA).

## Generation of GUS reporter constructs

The *CrLFY1*$_{pro}$*::GUS* reporter construct (*Figure 4—figure supplement 1*) was created by cloning the *CrLFY1* promoter into pART7 as a *Not*I-*Xba*I restriction fragment, replacing the existing *35S* promoter. A β-Glucuronidase (GUS) coding sequence (*Ulmasov, 1997*) was cloned downstream of *pCrLFY1* as an *Xba*I-*Xba*I fragment. The same GUS *Xba*I-*Xba*I fragment was also cloned into pART7 to create a *35S*$_{pro}$*::GUS* positive control (*Figure 4—figure supplement 4*). The resulting *CrLFY1*$_{pro}$*::GUS::ocs* and *35S*$_{pro}$*::GUS::ocs* cassettes were each cloned as *Not*I-*Not*I fragments into the pART27-based binary transformation vector pBOMBER carrying a hygromycin resistance marker previously optimized for Ceratopteris transformation (*Plackett et al., 2015*). All primers used in GUS reporter component amplification are supplied in the Key Resources table.

## Generation of RNAi constructs

RNAi constructs were designed and constructed using the pANDA RNAi expression system (*Miki and Shimamoto, 2004*). Four RNAi fragments were designed, two targeting a conserved region of the *CrLFY1* and *CrLFY2* coding sequence (77% nucleotide identity) using sequences from either *CrLFY1* (*CrLFY1/2-i1*) or *CrLFY2* (*CrLFY1/2-i2*), and two targeting gene-specific sequence within the 3'UTR of *CrLFY1* (*CrLFY1-i3*) or *CrLFY2* (*CrLFY2-i4*) (*Figure 6—figure supplement 1*). Target fragments were amplified from cDNA and cloned into Gateway-compatible entry vector pDONR207 (Invitrogen, Carlsbad, CA). Each sequence was then recombined into the pANDA expression vector via Gateway LR cloning (Invitrogen, Carlsbad, CA). All primers used in RNAi target fragment amplification are supplied in the Key Resources table.

## Generation of transgenic lines

Transformation of all transgenes into wild-type Hn-n Ceratopteris callus was performed as previously described (*Plackett et al., 2015*). $T_0$ sporophyte shoots were regenerated from transformed callus tissue, with each round of transformation using multiple separate pieces of callus as starting material. Transgenic $T_1$ spores were harvested from these $T_0$ shoots, germinated to form $T_1$ gametophytes and then self-fertilized to produce $T_1$ sporophytes. $T_1$ sporophytes were assessed for T-DNA copy number by DNA gel blot analysis (*Figure 4—figure supplement 2*; *Figure 6—figure supplement 3*) and the presence of full-length T-DNA insertions was confirmed through genotyping PCR (*Figure 4—figure supplements 3* and *4*). All primers used in genotyping reactions are supplied in the Key Resources table. For characterization of RNAi lines, $T_2$ spores were collected from individuals that either contained the full transgene construct or from segregants in which the RNAi hairpin was absent.

## GUS staining

GUS activity analysis in *CrLFY1$_{pro}$::GUS* transgenic lines was conducted in the $T_1$ generation. GUS staining was conducted as described previously (*Plackett et al., 2014*). Optimum staining conditions (1 mg/ml X-GlcA, 5 µM potassium ferricyanide) were determined empirically. Tissue was cleared with sequential incubations in 70% ethanol until no further decolorization occurred. GUS-stained gametophytes were imaged with a Zeiss Axioplan microscope and GUS-stained sporophytes imaged with a dissecting microscope, both mounted with Q-imaging Micro-published 3.3 RTV cameras. Images were minimally processed for brightness and contrast in Photoshop (CS4).

## Phenotypic characterization

Phenotypic characterization of RNAi transgenic lines was conducted in the $T_2$ or $T_3$ generation. Isogenic lines were obtained by isolating hermaphrodite gametophytes in individual wells at approximately 7 DPS (or when the notch became visible, whichever came first) and flooding them once they had developed mature gametangia (at approximately 9 DPS). All transgenic lines were grown alongside both wild-type and no hairpin controls, and phenotypes observed and recorded daily. Gametophytes exhibiting altered phenotypes were imaged at approximately 10 DPS with a Nikon Microphot-FX microscope. Sporophytes with abnormal phenotypes were imaged with a dissecting microscope.

## *In situ* hybridization

Antisense and sense RNA probes for *CrLFY1* and *CrLFY2* were amplified and cloned into pCR 4-TOPO (Invitrogen) and DIG-labelled according to the manufacturer's instructions (Roche, Indianapolis, IN). Probes were designed to include the 5'UTR and ORF (*CrLFY1* 521 bp 5'UTR and 1113 bp ORF; *CrLFY2* 301 bp 5'UTR and 1185 bp ORF) (*Supplementary file 6*). All primers used in *in situ* probe amplification are supplied in the Key Resources table. We were unable to identify fragments that distinguished the two genes in whole mount in situ hybridizations. Tissue was fixed in FAA (3.7% formaldehyde, 5% acetic acid; 50% ethanol) for 1–4 hr and then stored in 70% ethanol. Whole mount *in situ* hybridization was carried out based on *Hejátko et al. (2006)*, with the following modifications: hybridization and wash steps were carried out in 24-well plates with custom-made transfer baskets (0.5 mL microcentrifuge tubes and 30 µm nylon mesh, Small Parts Inc., Logansport, IN). Permeabilization and post-fixation steps were omitted depending on tissue type to avoid damaging

fragile gametophytes, Acetic Anhydride (Sigma-Aldrich) and 0.5% Blocking Reagent (Roche) washing steps were added to decrease background staining, and tissue was hybridized at 45°C. Photos were taken under bright-field with a Q-imaging Micro-publisher 3.3 RTV camera mounted on a Nikon Microphot-FX microscope. Images were minimally processed for brightness and contrast in Photoshop (CS4).

## Acknowledgements

Work in JAL's lab was funded by an ERC Advanced Investigator Grant (EDIP) and by the Gatsby Charitable Foundation. VSD's visit to JAL's lab was funded in part by NSF/EDEN IOS 0955517. Work in VSD's lab was funded by the Royalty Research Fund (A96605) and Bridge Funding Program (652315), University of Washington. We thank Brittany Dean, Henry Blazina and Jonathan Yee for their contribution to transgenic line screening and validation. We are grateful to Peng Wang for the primers PW64F and PW64R.

## Additional information

### Funding

| Funder | Grant reference number | Author |
|---|---|---|
| Horizon 2020 Framework Programme | ERC AdG EDIP | Jane A Langdale |
| Gatsby Charitable Foundation | | Jane A Langdale |
| National Science Foundation | EDEN IOS 0955517 | Verónica S Di Stilio |
| University of Washington | Royalty Research Fund, A96605 | Verónica S Di Stilio |
| University of Washington | Bridge Funding Program 652315 | Verónica S Di Stilio |

The funders had no role in study design, data collection and interpretation, or the decision to submit the work for publication.

### Author contributions

Andrew RG Plackett, Data curation, Formal analysis, Validation, Investigation, Visualization, Methodology, Writing—original draft, Cloned open reading frames and upstream promoter fragments of CrLFY1 and CrLFY2, Made the CrLFY1pro::GUS reporter constructs, Generated and validated transgenic reporter lines, Conducted GUS staining, Performed gel blot analysis of CrLFY copy number; Stephanie J Conway, Formal analysis, Validation, Investigation, Visualization, Methodology, Writing—original draft, Screened, validated and characterized RNAi lines, Carried out in situ hybridizations; Kristen D Hewett Hazelton, Formal analysis, Validation, Investigation, Methodology, Writing—review and editing, Screened, validated and characterized RNAi lines, Performed ontogenetic gene expression analysis; Ester H Rabbinowitsch, Investigation, Generated and maintained T0 transgenic lines; Jane A Langdale, Conceptualization, Formal analysis, Supervision, Funding acquisition, Validation, Investigation, Visualization, Writing—original draft, Carried out the phylogenetic analyses; Verónica S Di Stilio, Conceptualization, Formal analysis, Supervision, Funding acquisition, Validation, Investigation, Visualization, Writing—original draft, Cloned the CrLFY coding sequences and made the RNAi constructs during a sabbatical visit to the University of Oxford, Carried out the statistical analyses

### Author ORCIDs

Andrew RG Plackett (iD) https://orcid.org/0000-0002-2321-7849
Jane A Langdale (iD) https://orcid.org/0000-0001-7648-3924
Verónica S Di Stilio (iD) https://orcid.org/0000-0002-6921-3018

Decision letter and Author response
Decision letter https://doi.org/10.7554/eLife.39625.037
Author response https://doi.org/10.7554/eLife.39625.038

# Additional files

**Supplementary files**

• Supplementary file 1. LFY sequences included in phylogenetic analysis (in addition to *Sayou et al., 2014*) dataset).
DOI: https://doi.org/10.7554/eLife.39625.027

• Supplementary file 2. Alignment of all 120 LFY amino acid sequences used in phylogenetic analysis.
DOI: https://doi.org/10.7554/eLife.39625.028

• Supplementary file 3. Alignment of LFY amino acid sequences used in phylogenetic analysis (ferns only).
DOI: https://doi.org/10.7554/eLife.39625.029

• Supplementary file 4. Statistical comparison of *CrLFY* transcript levels between different ontogenetic stages.
DOI: https://doi.org/10.7554/eLife.39625.030

• Supplementary file 5. Specificity of *CrLFY* RNAi target sequences. Alignment (prepared using Clustal Omega) of the full length transcript sequences for *CrLFY1* and *CrLFY2*, with nucleotide identity between the two gene copies denoted by a subtending asterisk. The CDS for each gene is highlighted in bold. The target sequences for each RNAi construct are highlighted in light blue (*CrLFY1/2-i1*), light green (*CrLFY1/2-i2*), dark blue (*CrLFY1-i3*) or dark green (*CrLFY2-i4*). The *CrLFY1/2-i1* and *CrLFY1/2-i2* target sequences each have 77% similarity to the opposing gene transcript (BLAST2n, discontiguous megablast for highly similar sequences). The full-length *CrLFY1-i3* and *CrLFY2-i4* target sequences do not demonstrate significant sequence similarity to the opposing gene transcript or 3'UTR alone (BLAST2n, blastn for somewhat similar sequences) but short regions of similarity within the target sequence might explain the cross-reactivity observed.
DOI: https://doi.org/10.7554/eLife.39625.031

• Supplementary file 6. Predicted hybridization and specificity of *CrLFY in situ* hybridization probes. Alignment (prepared using Clustal Omega) of full length *CrLFY1* and *CrLFY2* transcript sequences, with nucleotide identity between the two paralogs denoted by a subtending asterisk. The coding sequence (CDS) for each gene copy is highlighted in bold. Predicted sites of hybridization for the two probes are highlighted in blue (*CrLFY1*) and yellow (*CrLFY2*) respectively, with PCR primer sites underlined. The in situ probes span the complete CDS and 5'UTR of each gene copy. The *CrLFY1* probe sequence shows 79% nucleotide identity to the *CrLFY2* transcript (BLAST2n, discontiguous megablast for highly similar sequences). The *CrLFY2* probe shows 79% nucleotide identity to the *CrLFY1* transcript (BLAST2n, discontiguous megablast for highly similar sequences).
DOI: https://doi.org/10.7554/eLife.39625.032

• Supplementary file 7. Amplified *CrLFY2* genomic fragment (3619 bp), not connected directly to *CrLFY2* open reading frame. Sequence highlighted in green corresponds to published *CrLFY2* 5'UTR (*Himi et al., 2001*)
DOI: https://doi.org/10.7554/eLife.39625.033

• Supplementary file 8. Summary of published reports of LFY function in a range of angiosperm species. All citations included in reference list of main article (*Bradley et al., 1996*; *Blázquez et al., 1997*; *Bradley et al., 1997*; *Kyozuka et al., 1998*; *Pnueli et al., 1998*; *Ratcliffe et al., 1999*; *Gourlay et al., 2000*; *Bomblies et al., 2003*; *Becker et al., 2005*; *Meng et al., 2007*; *Souer et al., 2008*; *Wreath et al., 2013*).
DOI: https://doi.org/10.7554/eLife.39625.034

• Transparent reporting form
DOI: https://doi.org/10.7554/eLife.39625.035

## Data availability

All data generated or analysed during this study are included in the manuscript and supporting files. Source data files have been provided for Figures 3 and 6. Sequences and alignments for phylogenetic analyses are included in Supplementary files 1-3.

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
