## [Decision Letter]

[Editors’ note: this article was originally rejected after discussions between the reviewers, but the authors were invited to resubmit after an appeal against the decision.]

Thank you for submitting your work entitled "*LEAFY* maintains apical stem cell activity during shoot development in the fern *Ceratopteris richardii*" for consideration by *eLife*. Your article has been reviewed by three peer reviewers, and the evaluation has been overseen by a Reviewing Editor and a Senior Editor. The reviewers have opted to remain anonymous.

Our decision has been reached after consultation between the reviewers. Based on these discussions and the individual reviews below, we regret to inform you that your work will not be considered for publication in *eLife* in its present form.

The reviewers agreed that *CrLFY2* promoter analysis and/or in situ hybridization, as well as controls for the RNAi analysis, are needed. *eLife* recommends a "reject" decision if it is likely that additional experiments will take more than 2 months. We therefore collectively agreed that rejection was the correct decision – however, the reviewers were all enthusiastic about the work and would strongly support re-submission to *eLife* if the requested experiments were included.

Reviewer #1:

This manuscript is the first to examine gene function using reverse genetics in a stably transformed fern, which is very exciting. The authors focus on *LEAFY*, which has been well characterized in moss and Arabidopsis. While it was previously known that the two Ceratopteris *LEAFY* genes are expressed in gametophytes and sporophytes, this study examines more detailed expression patterns and addresses their functions in both Ceratopteris generations by RNAi. The topic is very interesting and important in understanding fern development and plant evolution in general.

Comments/questions relating to the gene expression studies:

1) Please describe the *LFY* promoter more; e.g., does it include all sequences upstream of the a) transcript initiation site b) the translation start site c) second exon that would include the first intron? This may be in the details of the supplementary figures (I couldn't tell), but I think it's important to include such details in the text.

2) Any reason why *LFY2* promoter-GUS expression wasn't included? The conclusion that two *LFY* homologs in Ceratopteris function together to control of apical cell identity would be better supported if the expression patterns of *CrLFY2* were examined using the same strategy (GUS reporter).

3) I assume that a T_1_ sporophyte is the product a self of a gametophyte that came from the spores of a transformed sporophyte that was selected then regenerated from bombarded callus. Is this correct?

My comments/questions relating to the RNAi lines:

1) What does phenotypic screening in the line "phenotypic screening identified 10 lines with similar developmental defects that were associated with reduced *CrLFY* expression" mean? My concern here is that the phenotypes described may be caused by the callus derived shoot regeneration process, which typically happens during callus regeneration in many different plant species. The authors need to carefully characterize the phenotypes of multiple individual transgenic plants together with their segregated siblings that do not contain the RNAi construct. I would also suggest performing vigorous genetic analyses to clearly define and confirm the phenotypes from multiple independent transgenic lines (i.e., backcrossing to or by wild type and examining whether the phenotype is tightly correlated to the repression of *LFY* genes in the F2 population). Wouldn't selection for drug resistance be useful here?

2) If some of the lines never develop gametophytes with gametangia, how were T_2_ or T_3_ lines generated?

My comments regarding the Discussion:

Much of this study is based on a single fern (Ceratopteris) and on one gene (*LFY*). For this reason, I regard the discussion of the evolutionary trajectory of *LFY* to be very speculative. The last sentence "if a pathway for regulating stem cell activity was co-opted from the gametophyte into the sporophyte, it was the *LFY* pathway" should be changed to.. if a pathway for regulating stem cell activity was co-opted from the gametophyte into the sporophyte, the pathway involving *LFY* is more likely than the *KNOX* pathway." There could be other pathways as well.

Because the Discussion focuses on evolution, it is important that this study include at least some supporting data from at least one other fern.

Reviewer #2:

Comparative analyses between bryophytes and angiosperms have been provided the information how gene function evolved to facilitate developmental innovations during land plant evolution as the authors pointed out. Ferns are one of missing link though it takes very important position during land plant evolution. Ceratopteris has been recognized as a model fern because of the early life cycle. The establishment of transformation technique in Ceratopteris became a breakthrough in this field (Muthukumar et al., 2013, Plant Physiol.; Plackett et al., 2014; Bui et al., 2015, BMC Res. Notes). *LFY* is one of well-studied genes among land plant lineages (e.g. Maizel et al., 2005; Sayou et al., 2014). Functional analysis of fern Ceratopteris *LFY* genes should be interesting topic for many readers.

The authors carefully examined the expression level of two Ceratopteris *LFY* genes, *CrLFY1* and *2*, spatial expression analysis of *CrLFY1* using promoter analysis, and suppression analysis of *CrLFY1* and *CrLFY2* activity by RNAi. This paper will be one of model case of EvoDevo study using the fern Ceratopteris. The quality of analyses and figures is excellent, however two questions remain.

1) Why the authors did not provide the expression data by *CrLFY2* promoter analysis nor in situ hybridization data of *CrLFY2*.

The authors indicated that *CrLFY1* and *CrLFY2* were differentially expressed during the Ceratopteris life cycle (subsection “*CrLFY1* and *CrLFY2* transcripts accumulate differentially during the Ceratopteris lifecycle”) and *CrLFY1* and *CrLFY2* act partially redundantly to maintain indeterminacy of the shoot apex in Ceratopteris (subsection “*CrLFY1* regulates activity of the sporophyte shoot apex”). Also they discussed that the duplicate *CrLFY* genes therefore act at least partially redundantly during shoot development in Ceratopteris (Discussion, first paragraph). It would be very helpful to understand the difference and redundancy in the function of *CrLFY1* and *2*, if they have the spatial expression data of *CrLFY2*.

The authors described that no GUS activity was detected in unfertilized archegonia of *CrLFY1_pro_::GUS* gametophytes (subsection “Spatial expression patterns of *CrLFY1* are consistent with a retained ancestral role to facilitate cell divisions during embryogenesis”). In the case, the expression of *CrLFY2* should to be important to function in the gametophytes. The spatial expression patterns of *CrLFY2* would provide important information for the hypothesis. Since the authors succeeded both transformation and in situ hybridization in Ceratopteris, it seems technically possible to provide these data.

2) The authors provided that RNAi suppression analysis of *CrLFY1* and *CrLFY2* by using 4 different constructs and used wild type strain as control. The strain introduced with the vector is better to use as control. Transformation trials from three constructs produced a few transformants and the phenotype from same construct is different. The authors explained the difference of phenotype was because of the expression level of *CrLFY1* or *CrLFY2*, though also should consider the effect of transformation procedure or transformation with the vector per se.

Reviewer #3:

This manuscript analyses the role of the *LEAFY* transcription factor in the fern Ceratopteris. The major finding is that *LEAFY* is expressed and essential for apical cell division both in the gametophyte and the sporophyte. *LEAFY* is a transcription factor playing a major role during flower development in angiosperms. It is present in most charophytes algae and all land plants. Its role has only been analyzed genetically in the moss *Physcomitrella patens* where it controls cell division in the sporophyte. In angiosperms, there is a growing body of evidence that *LEAFY* plays a role in meristem development (not necessarily flower meristem) as illustrated in various species such as rice, cucumber or pea. The evolutionary scenario describing how *LEAFY* could have been co-opted from a role in sporophyte cell division to flower development has remained very uncertain because of the lack of genetics in plant groups such as ferns and gymnosperms. In species belonging to these groups, one had to rely on expression patterns and inference on the function but there was, until now, no genetic evidence.

The work presented here represents a great leap forward by partially filling the gap between mosses and angiosperms. It analyses the expression and function of the *LEAFY* genes in Ceratopteris and concludes that *LEAFY* plays a role in apical growth in the sporophyte and gametophyte. To my opinion, these conclusions are well supported. Generating and analysing fern transgenics is a challenging but extremely useful task to learn about plant evolution in the major plant families. Even if the work is based mostly on expression pattern and mutant description (and it might look a bit old-fashioned at first glance for the lack of attempt to identify regulated genes), I am convinced that the results presented here are of great significance not only for the *LEAFY* research field but as an example of how a transcription factor can be co-opted along plant evolution to acquire a key role in angiosperms. Such a trajectory could not have been guessed without this type of seminal work in more basal plants.

My only concern is that this work is not so easy to follow for those who are not familiar with fern development. I suggest adding a scheme describing Ceraptopteris life cycle (and naming relevant organs and tissues) that can referred to as a guide for each figure showing expression patterns (GUS or mRNA) or phenotypes.

---

## [Author Response]

[Editors’ note: the author responses to the first round of peer review follow.]

Many thanks for the well-considered handling of our manuscript. The reviews and decision are perfectly understandable based on the contents of the submitted manuscript and I am grateful to the reviewers for their enthusiasm and informed critique. I am especially grateful for the acceptance that any further experiments would be impossible in two months. However, I attach a document that provides some extra information relating to the two main issues – *CrLFY2* promoter analysis and RNAi controls. I have highlighted the main points where we can add information to clarify/justify our approach. I do believe that the conclusions we have made are robust and justifiable but accept that we could be more circumspect in the discussion.

In brief:

1) The *CrLFY2* analysis is impossible until a genome sequence is available.

2) We chose wild-type as the standard control for RNAi lines because null segregants are rarely that after bombardment in Ceratopteris – each is likely to contain a fragment of the construct even if the hairpin is absent.

Reviewer #1:[…] Comments/questions relating to the gene expression studies:1) Please describe the LFY promoter more; e.g., does it include all sequences upstream of the a) transcript initiation site b) the translation start site c) second exon that would include the first intron? This may be in the details of the supplementary figures (I couldn't tell), but I think it's important to include such details in the text.

A diagrammatic representation of the promoter and 5’UTR structure was included as part of Figure 1—figure supplement 2 and referred to in the Materials and methods. Details are now also in the

text of the Results.

2) Any reason why LFY2 promoter-GUS expression wasn't included? The conclusion that two LFY homologs in Ceratopteris function together to control of apical cell identity would be better supported if the expression patterns of CrLFY2 were examined using the same strategy (GUS reporter).

It would have been ideal to include expression data for both genes and numerous attempts were made to obtain a *CrLFY2* promoter fragment for use in GUS expression studies alongside *CrLFY1*, but this proved impossible in the absence of a reference genome. Details of what we attempted to do are now in the Materials and methods, and reference to the fact that we were unable to amplify a *CrLFY2* promoter for use in GUS assays is now in the results text. We also refer to the fact that expression could not be determined in situ as we were unable to identify a fragment that clearly distinguished *CrLFY1* and *CrLFY2* in hybridizations (in main text and Materials and methods).

3) I assume that a T_1_ sporophyte is the product a self of a gametophyte that came from the spores of a transformed sporophyte that was selected then regenerated from bombarded callus. Is this correct?

This is now clearly articulated in the Materials and methods.

My comments/questions relating to the RNAi lines:1) What does phenotypic screening in the line "phenotypic screening identified 10 lines with similar developmental defects that were associated with reduced CrLFY expression" mean? My concern here is that the phenotypes described may be caused by the callus derived shoot regeneration process, which typically happens during callus regeneration in many different plant species. The authors need to carefully characterize the phenotypes of multiple individual transgenic plants together with their segregated siblings that do not contain the RNAi construct.

We compared the phenotype of transgenic lines containing the full transgene with that of wild-type and of transgenic controls that lacked the RNAi hairpin (no hairpin controls – NHC). Wild type and NHC lines were phenotypically indistinguishable and NHC lines did not show suppression of endogenous *LFY* expression (see new Figure 6, Supplementary file 6, Table 1). Because DNA fragmentation after bombardment is common, these lines still contain part of the transgene cassette. Importantly, the new controls do not change the interpretation of any of the RNAi data.

I would also suggest performing vigorous genetic analyses to clearly define and confirm the phenotypes from multiple independent transgenic lines (i.e., backcrossing to or by wild type and examining whether the phenotype is tightly correlated to the repression of LFY genes in the F2 population). Wouldn't selection for drug resistance be useful here?

We did not do this routinely but because the selection antibiotic prevented fertilization of gametophytes, we did in some cases grow T_1_ gametophyte populations on non-selective medium and allow unrestricted mating to occur within lines which were determined to be a mixed population of transgenic and non-transgenic individuals (based on relative survival on and off antibiotic selection). Transgenic individuals were then identified post-fertilization by screening directly for antibiotic resistance. In these cases, T_1_ sporophytes would represent either hemizygous or homozygous individuals, some of the hemizygous being effectively backcrossed to wild-type. Gel-blot analysis found that sibling sporophytes generated within a line in this manner had a similar pattern of T-DNA insertions, implying very little genetic variation within each T_1_ population. There was also very little phenotypic variation between individuals.

2) If some of the lines never develop gametophytes with gametangia, how were T_2_ or T_3_ lines generated?

There were only two lines in which 100% of the gametophytes arrested (F14 and E8). It was thus possible to look at sporophyte development in the remaining 8 (7 of which are shown in new Figure 6).

My comments regarding the Discussion:Much of this study is based on a single fern (Ceratopteris) and on one gene (LFY). For this reason, I regard the discussion of the evolutionary trajectory of LFY to be very speculative. The last sentence "if a pathway for regulating stem cell activity was co-opted from the gametophyte into the sporophyte, it was the LFY pathway" should be changed to.. if a pathway for regulating stem cell activity was co-opted from the gametophyte into the sporophyte, the pathway involving LFY is more likely than the KNOX pathway." There could be other pathways as well.

This sentence has been changed.

Because the Discussion focuses on evolution, it is important that this study include at least some supporting data from at least one other fern.

This is not feasible at this stage because of the paucity of other model fern systems, neither has it been a requirement in other evo-devo studies e.g., with mosses (focused on *Physcomitrella patens*) or liverworts (*Marchantia polymorpha*). The point of the Discussion is to set up a hypothesis that can be further validated in future studies.

Reviewer #2:[…] The authors carefully examined the expression level of two Ceratopteris LFY genes, CrLFY1 and 2, spatial expression analysis of CrLFY1 using promoter analysis, and suppression analysis of CrLFY1 and CrLFY2 activity by RNAi. This paper will be one of model case of EvoDevo study using the fern Ceratopteris. The quality of analyses and figures is excellent, however two questions remain.1) Why the authors did not provide the expression data by CrLFY2 promoter analysis nor in situ hybridization data of CrLFY2.The authors indicated that CrLFY1 and CrLFY2 were differentially expressed during the Ceratopteris life cycle (subsection “CrLFY1 and CrLFY2 transcripts accumulate differentially during the Ceratopteris lifecycle”) and CrLFY1 and CrLFY2 act partially redundantly to maintain indeterminacy of the shoot apex in Ceratopteris (subsection “CrLFY1 regulates activity of the sporophyte shoot apex”). Also they discussed that the duplicate CrLFY genes therefore act at least partially redundantly during shoot development in Ceratopteris (Discussion, first paragraph). It would be very helpful to understand the difference and redundancy in the function of CrLFY1 and 2, if they have the spatial expression data of CrLFY2.The authors described that no GUS activity was detected in unfertilized archegonia of CrLFY1_pro_::GUS gametophytes (subsection “Spatial expression patterns of CrLFY1 are consistent with a retained ancestral role to facilitate cell divisions during embryogenesis”). In the case, the expression of CrLFY2 should to be important to function in the gametophytes. The spatial expression patterns of CrLFY2 would provide important information for the hypothesis. Since the authors succeeded both transformation and in situ hybridization in Ceratopteris, it seems technically possible to provide these data.

See response 2 to reviewer 1.

2) The authors provided that RNAi suppression analysis of CrLFY1 and CrLFY2 by using 4 different constructs and used wild type strain as control. The strain introduced with the vector is better to use as control. Transformation trials from three constructs produced a few transformants and the phenotype from same construct is different. The authors explained the difference of phenotype was because of the expression level of CrLFY1 or CrLFY2, though also should consider the effect of transformation procedure or transformation with the vector per se.

See response 4 to reviewer 1.

Reviewer #3:[…] My only concern is that this work is not so easy to follow for those who are not familiar with fern development. I suggest adding a scheme describing Ceraptopteris life cycle (and naming relevant organs and tissues) that can referred to as a guide for each figure showing expression patterns (GUS or mRNA) or phenotypes.

We have inserted a schematic of the lifecycle showing morphology at different stages as new Figure 2. Citations of studies describing details of Ceratopteris development and morphology have now also been included in the text where appropriate.